# Inhibition of IRF4 in dendritic cells by PRR-independent and -dependent signals inhibit Th2 and promote Th17 responses

Jihyung Lee[1], Junyan Zhang[1,2,3], Young-Jun Chung[1,4], Jun Hwan Kim[1], Chae Min Kook[1], José M González-Navajas[3,5,6], David S Herdman[1], Bernd Nürnberg[7], Paul A Insel[1,8], Maripat Corr[1], Ji-Hun Mo[4], Ailin Tao[2,3], Kei Yasuda[9], Ian R Rifkin[9,10], David H Broide[1], Roger Sciammas[11], Nicholas JG Webster[1,12]*, Eyal Raz[1,3]*

[1]Department of Medicine, University of California San Diego, San Diego, United States; [2]The Second Affiliated Hospital of Guangzhou Medical University (GMU), The State Key Laboratory of Respiratory Disease, Guangdong Provincial Key Laboratory of Allergy & Clinical Immunology, Guangzhou, China; [3]Center for Immunology, Inflammation and Immune-mediated disease, GMU, Guangzhou, China; [4]Department of Otorhinolaryngology-Head and Neck Surgery, Dankook University College of Medicine, Chungnam, Republic of Korea; [5]Alicante Institute for Health and Biomedical Research (ISABIAL - FISABIO), Alicante, Spain; [6]Networked Biomedical Research Center for Hepatic and Digestive Diseases (CIBERehd), Institute of Health Carlos III, Madrid, Spain; [7]Department of Pharmacology and Experimental Therapy, University of Tübingen, Tübingen, Germany; [8]Department of Pharmacology, University of California San Diego, San Diego, United States; [9]Boston University School of Medicine, Boston, United States; [10]VA Boston Healthcare System, Boston, United States; [11]Center for Comparative Medicine, University of California, Davis, Davis, United States; [12]VA San Diego Healthcare System, San Diego, United States

*For correspondence:
nwebster@ucsd.edu (NJGW);
eraz@ucsd.edu (ER)

Competing interests: The authors declare that no competing interests exist.

**Abstract** Cyclic AMP (cAMP) is involved in many biological processes but little is known regarding its role in shaping immunity. Here we show that cAMP-PKA-CREB signaling (a pattern recognition receptor [PRR]-independent mechanism) regulates conventional type-2 Dendritic Cells (cDC2s) in mice and reprograms their Th17-inducing properties via repression of IRF4 and KLF4, transcription factors essential for cDC2-mediated Th2 induction. In mice, genetic loss of IRF4 phenocopies the effects of cAMP on Th17 induction and restoration of IRF4 prevents the cAMP effect. Moreover, curdlan, a PRR-dependent microbial product, activates CREB and represses IRF4 and KLF4, resulting in a pro-Th17 phenotype of cDC2s. These in vitro and in vivo results define a novel signaling pathway by which cDC2s display plasticity and provide a new molecular basis for the classification of novel cDC2 and cDC17 subsets. The findings also reveal that repressing IRF4 and KLF4 pathway can be harnessed for immuno-regulation.

## Introduction

Pattern recognition receptors (PRRs) are germline-encoded proteins expressed primarily on innate immune cells that recognize conserved microbe- or pathogen-associated molecular patterns (PAMPs), and damage-associated molecular pattern (DAMPs). The main paradigm of immune activation posits that triggering of PRRs results in the maturation of DCs and their subsequent activation-

acquired properties to elicit and shape CD8$^+$ T cells and CD4$^+$ Th cells responses (*Hansen et al., 2011*; *Palm and Medzhitov, 2009*; *Takeuchi and Akira, 2010*). DCs, the main antigen presenting cells (APCs), display migratory and functional heterogeneity, and are distributed throughout the body (*Guilliams et al., 2014*). Conventional DCs (cDCs) are found in most tissues and thought to be of two major lineages, each of which expresses a distinctive set of surface receptors and a unique set of TFs that regulate development and function of DCs (*Guilliams et al., 2014*; *Murphy, 2013*). For example, the splenic cDC1 subpopulation, which promotes CD4$^+$ Th1 and CD8$^+$ cytotoxic T cells (CTL) responses, requires the TFs interferon regulatory factor 8 (IRF8) and basic leucine-zipper ATF-like transcription factor 3 (BATF3), whereas splenic cDC2s, which promote Th2 and Th17 responses, require IRF4 (*Gao et al., 2013*; *Hambleton et al., 2011*; *Vander Lugt et al., 2014*; *Williams et al., 2013*) and Kruppel-like factor 4 (KLF4) (*Tussiwand et al., 2015*). The cDC2s lineage has been further divided into Th2-inducing (IRF4$^+$/KLF4$^+$), and Th17-inducing (IRF4$^+$/NOTCH2$^+$) subpopulations (*Bedoui and Heath, 2015*; *Tussiwand et al., 2015*). These cDCs express many PRRs to trigger DC maturation but can elicit and shape the CD4$^+$ Th response in the absence of such stimulation. DCs also express multiple G protein-coupled receptors (GPCRs) but their role, in particular of GPCRs that regulate cAMP formation, in DC-related innate immunity and hence, on adaptive immunity, is poorly understood (*Idzko et al., 2014*; *Pearce and Everts, 2015*). Thus, even though cAMP regulates many biological processes, its impact on immune responses is not well defined.

IRF4 influences B and T lymphocyte antigen-dependent responses by controlling the effector properties of expanded clones (*De Silva et al., 2012*; *Huber and Lohoff, 2014*). Induced as an immediate early gene by antigen receptors, IRF4's expression scales with the intensity of receptor signaling, thus linking the quality of antigen receptor signaling with B and T cell fate output (*Krishnamoorthy et al., 2017*; *Man et al., 2013*; *Matsuyama et al., 1995*; *Nayar et al., 2014*; *Ochiai et al., 2013*; *Sciammas et al., 2006*; *Yao et al., 2013*). Differing concentrations of IRF4 induced by antigen receptor signaling are thought to promote differential assembly of IRF4 into distinct TFs and DNA recognition complexes to regulate gene expression programs important for B and T cell fate (*Krishnamoorthy et al., 2017*; *Ochiai et al., 2013*). In contrast to other members of the IRF family, expression of IRF4 is not induced by either Type I or Type II interferon (*Matsuyama et al., 1995*) but rather by different modes of NF-kB signaling induced by the antigen receptors, TLR, and TNF receptor systems (*Shaffer et al., 2009*). IRF4 plays important roles in controlling the stimulatory properties of DCs but the mechanisms and contexts by which these are deployed to regulate Th responses are less well understood (*Bajaña et al., 2012*; *Vander Lugt et al., 2014*).

We have found that low levels of cAMP in cDCs promote Th2 differentiation (*Lee et al., 2015*) and high levels promote induction of Th17 (*Datta et al., 2010*). In the current study, we sought to dissect the molecular mechanisms by which cAMP levels regulate these Th responses by cDC1 and cDC2 subsets and identified a previously unappreciated DC plasticity provoked by PRR-independent (cAMP) and PRR-dependent (curdlan) signaling that affects Th bias.

Overall, these findings implicate a previously unappreciated DC plasticity provoked by PRR-independent (cAMP) and PRR-dependent (curdlan) signaling that affects Th bias.

## Results

### Cyclic AMP signaling reprograms cDC2 cells from a pro-Th2 to a pro-Th17 phenotype

We previously reported that cAMP signaling in CD11c$^+$bone marrow derived DCs (BMDCs), that is, BM-derived antigen presenting cells (BM-APCs), affects the differentiation of CD4$^+$ T cells (*Datta et al., 2010*; *Lee et al., 2015*). Low cAMP levels, as occurs in the mice with deletion of *Gnas* in CD11c-expressing cells (Gnas$^{\Delta CD11c}$ mice, generated by breeding of *Gnas* floxed mice (Gnas$^{fl/fl}$) with CD11c-Cre mice), provoke a Th2 polarization that leads to an allergic phenotype (*Lee et al., 2015*), while cholera toxin (CT) and other treatments that increase cAMP levels in CD11c$^+$ cells, induce differentiation to Th17 cells (*Datta et al., 2010*). Given these observations, it was important to investigate how this phenotypic reprogramming occurs in *bona fide* DCs.

Splenic DCs have been divided into three subsets: two cDC subsets (cDC1s and cDC2s), and plasmacytoid DCs (pDCs) (*Guilliams et al., 2014*; *Sichien et al., 2017*). We tested the effect of the cell-

permeable cAMP analog 8-(4-Chlorophenylthio) adenosine 3',5'-cyclic monophosphate (CPT) on cDC2s and cDC1s. cDC2s (CD11c$^+$CD11b$^+$CD8α$^-$ splenocytes) (*Hey and O'Neill, 2012*) were isolated by FACS sorting, pulsed with MHC class II (MHCII) OVA peptide and co-cultured with naïve OVA-specific splenic OT-II CD$^{4+}$ T cells (OT-II cells). Treatment of WT cDC2s with CPT decreased IL-4 (*Figure 1A*) and increased IL-17A concentrations (*Figure 1B*) in co-cultured OT-II cells. IFN-γ and IL-10 concentrations were not changed by CPT-treated cDC2s (*Figure 1C,D*). Analysis of the T cell lineage commitment factors of OT-II cells co-cultured with CPT-treated cDC2s revealed a decrease in *Gata3* and increase in *Nr1f3* levels (*Figure 1E*). Induction of IL-17 by CPT-treated cDC2 was 2-fold greater in IL-17 GFP OT-II cells (*Figure 1F*). The TFs *Irf4* (*Gao et al., 2013*; *Williams et al., 2013*) and *Klf4* (*Tussiwand et al., 2015*) regulate the pro-Th2 phenotype of DCs. CPT decreased the expression of *Irf4* and *Klf4* in cDC2s in a time-dependent manner; *Irf8* expression was not affected. To verify that cAMP signaling was activated, we assessed expression of the cAMP-induced gene *Crem* and observed a time-dependent, 10-fold induction (*Figure 1G*). Decreased expression of IRF4 after CPT treatment was confirmed by intracellular FACS staining (*Figure 1H*). We also tested the effect of CPT on the transcriptional program of cDC1s, we isolated CD11c$^+$CD11b$^-$CD8α$^+$ splenocytes and co-cultured them with OT-II cells. Unlike what occurred with cDC2s, CPT treatment of cDC1s did not change the T cell cytokines or T cell lineage commitment factors in the co-cultured OT-II cells (*Figure 1—figure supplement 1A–D*). Furthermore, *Crem* was induced but expression of *Irf4* and *Klf4* was not changed by CPT treatment (*Figure 1—figure supplement 1E*). cDC2 cells had higher IRF4 (*Figure 1H*), IRF5 (*Figure 1—figure supplement 1F*) and lower IRF8 (*Figure 1—figure supplement 1G*) compared to cDC1s. Expression of IRF5 and IRF8 was not changed by CPT treatment (*Figure 1—figure supplement 1H,I*).

Memory Th cells are divided into two subsets: effector memory T (TEM, CD44$^+$CD62L$^{low}$) and central memory T (TCM, CD44$^+$CD62L$^{high}$) cells (*Nakayama et al., 2017*). To assess a possible switch in fate from Th2EM to Th17 cells (*Hirota et al., 2011*; *Nakayama et al., 2017*), we first generated Th2EM cells by co-culturing naïve IL-17GFP OT-II cells with Gnas$^{ΔCD11c}$ BM-APCs (*Lee et al., 2015*) and then sorted T1/ST2$^+$ cells (*Lohning et al., 1998*). More than 96% of T1/ST2$^+$ cells from the co-culture were Th2EM and FOXP3$^-$ (*Figure 1—figure supplement 2A*). The sorted cells were then used for a second co-culture with CPT- or cholera toxin (CT) (*Datta et al., 2010*)-treated WT cDC2s. Th2EM co-cultured with CPT- or CT-treated WT cDC2s had decreased expression of IL-4 (*Figure 1I*), increased IL-17A expression (*Figure 1J*) but unchanged expression of IFN-γ (*Figure 1K*). Th2EM co-cultured with CPT- or CT-treated cDC2s had altered expression of T cell lineage commitment factors: decreased expression of *Gata3* and increased expression of *Nr1f3* and *Nr1f1* (*Figure 1L*). CPT- or CT-induced Th17 differentiation was confirmed by GFP expression (FACS) in re-stimulated OT-II cells (*Figure 1M*). Expression of T1/ST2 and memory T cell markers, CD62L and CD44, was FOXP3$^-$ (*Figure 1—figure supplement 2B*) but not changed by the second co-culture (data not shown). Overall, these findings suggest that these cells are akin to Th2-Th17 hybrid cells (*Wang et al., 2010*).

To confirm these effects in a genetic model, we isolated CD11c$^+$CD11b$^+$CD8α$^-$ splenocytes from Gnas$^{fl/fl}$ and Gnas$^{ΔCD11c}$ mice (which have a prominent decrease in expression of the Gαs protein and in cAMP synthesis and as a result, a bias toward Th2 induction) (*Lee et al., 2015*) and tested the cDC2 in co-culture with OT-II cells. We found that cDC2s from Gnas$^{ΔCD11c}$ mice elicited a 9.7-fold greater IL-4 response, which was completely suppressed by CPT treatment (*Figure 1—figure supplement 3A*). IL-17A response induced by CPT-treated Gnas$^{ΔCD11c}$ cDC2s increased by 3.4-fold and 2.6-fold in CPT-treated Gnas$^{fl/fl}$ cDC2 (*Figure 1—figure supplement 3B*). Treatment with CPT suppressed *Gata3* and induced higher *Nr1f3* levels in co-cultured OT-II cells without altering *Tbx21* or *Foxp3* (*Figure 1—figure supplement 3D*). Basal *Irf4* and *Klf4* levels in cDC2s were elevated in the Gnas$^{ΔCD11c}$ compared to Gnas$^{fl/fl}$ cDC2 but were suppressed in response to CPT (*Figure 1—figure supplement 3E,F*). CPR treatment induced *Crem* but did not change *Irf8* expression (*Figure 1—figure supplement 3E*).

To confirm that BM-APCs are a valid model for DC reprogramming and to gain further insight into mechanisms underlying this process, we tested WT CD11c$^+$CD135$^+$ BM-APCs. Agonists that increase cAMP induced a pro-Th17 phenotype in BM-APCs (*Figure 1—figure supplement 4A,B*) and a 2-fold increase in *Nr1f3* expression (*Figure 1—figure supplement 4C*). Expression of *Irf4* and *Klf4* was also decreased by treatment of WT BM-APCs with agonists that increase cAMP (*Figure 1—figure supplement 4D*). We assessed the impact of three cAMP signaling effectors, PKA

(*Glass et al., 1989*), EPAC (*Parnell et al., 2015*), and CREB (*Xie et al., 2015*), in the inhibition of *Irf4* by PGE2, a Gαs-coupled GPCR agonist (*Wehbi and Taskén, 2016*). Treatment with Rp-cAMP (a PKA inhibitor) or 666–15 (a CREB inhibitor) but not with CE3F4(an EPAC inhibitor) abolished the PGE2-promoted reduction in *Irf4* expression (*Figure 1—figure supplement 4E,F*).

We also assessed the effect of cAMP treatment on human DC-like cell lines generated from MUTZ-3 (myelomonocyte) (*Mitra et al., 2013*), THP-1 (monocyte) (*Berges et al., 2005*), and HL-60 (promyeloblast) (*Koski et al., 1999*). As observed with mouse cDC2s or BM-APCs, the human DC-like cells responded to CPT with a decrease in expression of *Irf4* and *Klf4* (*Figure 2*).

Collectively, these results indicate that increased cAMP levels reprogram transcription of cDC2s and BM-APCs from WT and Gnas$^{\Delta CD11c}$ mice. Low cAMP concentrations promote expression of a pro-Th2 phenotype while high cAMP concentrations promote pro-Th17 inducing properties and a subsequent switch from Th2 to Th17 bias. Furthermore, high cAMP concentrations can override an existing pro-Th2 phenotype and reprogram cDC2 into pro-Th17-inducing cDC17s. This phenotypic switch in DC2s is mediated by cAMP signaling via PKA/CREB, but not EPAC, and is associated with the inhibition of *Irf4* and *Klf4*.

## PRR-dependent signaling down-regulates IRF4 and KLF4 expression and induces a pro-Th17 phenotype of DCs

To investigate if the reprogramming of WT cDC2 is specific for cAMP signaling, we tested the effect of a microbial Th17 inducer, curdlan. Curdlan is a linear polymer of β (1,3)-glucan derived from soil bacteria and the ligand of the PRR Dectin-1 (*Yoshitomi et al., 2005*). Signaling by Dectin-1 is independent of cAMP and involves recruitment of Syk to the Dectin-1 intracellular tail, followed by activation of MAPK, NF-κB and NFAT (*Gross et al., 2006*) and induction of IL-23, which promotes Th17 differentiation (*LeibundGut-Landmann et al., 2007*; *Rogers et al., 2005*). WT cDC2s express Dectin-1 (*Figure 3A*). Treatment of these cells with curdlan increased IL-17A concentrations 2.5-fold (*Figure 3B*) and *Nr1f3* mRNA levels 1.7-fold (*Figure 3C*) in co-cultured OT-II cells. Akin to what occurred with CPT treatment, curdlan inhibited the expression of *Irf4* and *Klf4*, but not *Irf8*, in cDC2s (*Figure 3D*) and induced *Il23a* (*Agrawal et al., 2010*) 6-fold (*Figure 3E*). The increase in IL-17A in curdlan-treated cDC2s was blocked by the CREB inhibitor, 666–15, but not by the PKA inhibitor, Rp-cAMP (*Figure 3F,G*). Unlike cDC2s, cDC1s cells do not express, or express very low levels of Dectin-1 (*Figure 3—figure supplement 1A*). Accordingly, curdlan treatment of cDC1s did not change the expression of T cell cytokines in co-cultured OT-II cells (*Figure 3—figure supplement 1B,C*), nor that of *Irf4* and *Il23a* in cDC1s (*Figure 3—figure supplement 1D,E*). Overall, these data indicate that curdlan acts via its PRR to induce a Th17 bias by cDC2s by a CREB-dependent, cAMP-PKA-independent pathway. Furthermore, the PRR-mediated and cAMP-mediated activation pathways converge on CREB to provide phenotypic plasticity of DCs, thus defining a new basis for classification of novel cDC2 and cDC17 subsets.

## Loss of IRF4 or IRF5 in DCs promotes or inhibits, respectively, the pro-Th17 DC phenotype and subsequent Th17 bias

IRF4 is a key TF in the development and function of innate immune cells (macrophages and cDCs) and adaptive immune cells (B and T cells) (*Biswas et al., 2012*; *Mittrücker et al., 1997*; *Satoh et al., 2010*; *Suzuki et al., 2004*). IRF4-deficient cDCs display dysfunctional antigen processing and presentation (*Vander Lugt et al., 2014*) and do not migrate to draining lymph nodes, which may contribute to their inability to stimulate Th2 or Th17 responses (*Bajaña et al., 2012*; *Vander Lugt et al., 2014*). The findings above indicate that cAMP signaling in splenic cDC2s inhibits IRF4 expression and provokes a Th17 bias. To determine if inhibition of IRF4 levels affects the pro-Th17 phenotype, we used splenic cDC2s from *Irf4*-inducible (*Irf4$^{-/-}$*) mice (*Ochiai et al., 2013*). These *Irf4$^{-/-}$* mice have a tetracycline (tet)-inducible cDNA allele of *Irf4* and the M2rtTA tet-activator allele, hereafter termed Irf4i mice. Doxycycline (Dox) induces IRF4 expression in cDC2s from these mice (*Figure 4A*). Unlike WT cDC2s, Irf4i cDC2s pulsed with MHCII OVA peptide (i.e., without any additional stimulation) increased IL-17A and decreased IL-5 production in co-cultured OT-II cells (*Figure 4B,C*). Activation of IRF4 with Dox treatment inhibited IL-17A production (*Figure 4B*), increased IL-5 levels (*Figure 4C*) but IL-4 was not detected and IFN-γ was not affected by the expression of IRF4 in these cDC2s (*Figure 4D*). Expression of the T cell lineage commitment factors

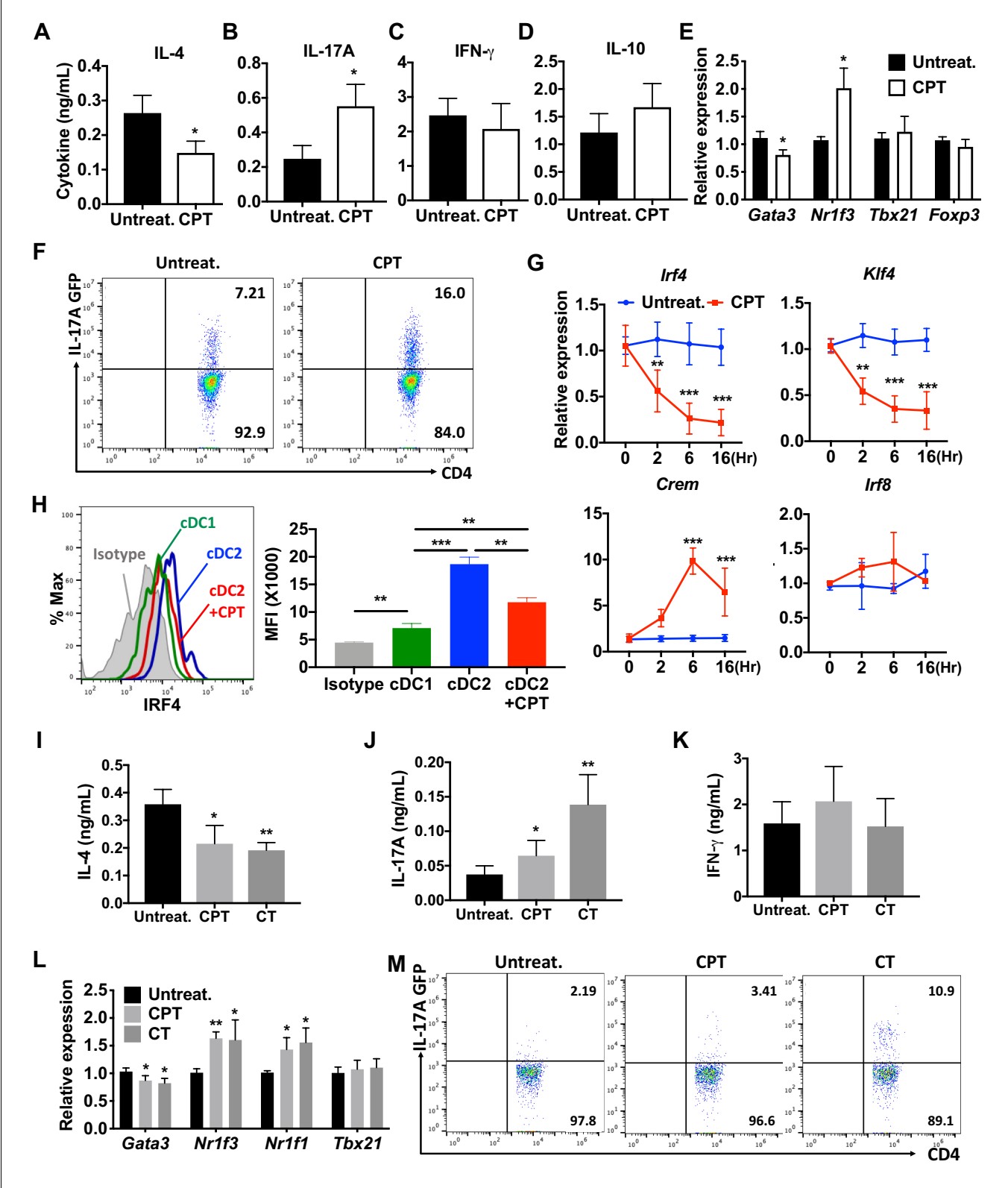

**Figure 1.** cAMP signaling switches cDC2s to a pro-Th17 bias. (A–D) IL-4, IL-17A, IFN-γ and IL-10 levels from anti-CD3/28 Ab-stimulated OT-II cells co-cultured with WT splenic cDC2s (CD11c$^+$CD11b$^+$CD8α$^-$) pretreated with or without CPT. (E) qPCR analysis of lineage commitment factors in OT-II T cells co-cultured with WT cDC2s in the presence of CPT. (F) GFP expression from IL-17GFP OT-II CD4$^+$ T cells co-cultured with WT cDC2s pretreated with or without CPT. (G) qPCR of TFs in WT cDC2s treated with CPT (50 µM). Two-way ANOVA with Sidak's multiple comparisons test; n = 3 in each
*Figure 1 continued on next page*

*Figure 1 continued*

group, \**p<0.01, \***p<0.001. Effect of CPT treatment; *Irf4* (p=0.002), *Klf4* (p<0.001) and *Crem* (p<0.001). (H) Intracellular staining of IRF4 in WT cDC1 and cDC2s treated with or without CPT for 48 hr. (I–M) Fate mapping: IL-17GFP OT-II CD4+ T cells were co-cultured with Gnas$^{\Delta CD11c}$ BM-APCs to generate memory Th2 cells (1st co-culture). From the 1st co-culture, T1/ST2+ cells were FACS sorted and then used for co-culture with WT cDC2s pretreated with or without CPT or Cholera toxin (CT) (2nd co-culture). (I) IL-4, (J) IL-17A and (K) IFN-γ levels, (L) qPCR of lineage commitment factors, and (M) GFP signal for IL-17 expression in the re-stimulated CD4+ T cells from 2nd co-culture. Data are representative of three independent experiments; \*p<0.05, \**p<0.01, \***p<0.001.

The online version of this article includes the following figure supplement(s) for figure 1:

**Figure supplement 1.** cAMP signaling in cDC1s does not affect the expression of IRF and subsequent T cell differentiation.

**Figure supplement 2.** FOXP3 expression in T1/ST2+ cells OT-II CD4+ T cells were co-cultured with Gnas$^{\Delta CD11c}$ BM-APCs to generate memory Th2 cells (1 st co-culture).

**Figure supplement 3.** cAMP signaling switches a pro-Th2 Gnas$^{\Delta CD11c}$ to a pro-Th17 phenotype.

**Figure supplement 4.** Induction of pro-Th17 BM-APCs and altered transcriptional program by cAMP agonists.

*Gata3* was increased and *Nr1f3* was decreased in OT-II cells co-cultured with Dox-treated Irf4i cDC2 (*Figure 4E*). Dox-treated BM-APCs from Irf4i mice also had increased IRF4 (*Figure 4—figure supplement 1A*). As with cDC2s, Irf4i BM-APCs induced Th17 differentiation spontaneously and restoring IRF4 expression with Dox treatment inhibited the increase in IL-17 (i.e., Th17) (*Figure 4—figure supplement 1B*). Moreover, restoration of IRF4 expression in BM-APCs induced Th2 differentiation; CPT did not change the restored IRF4 expression and subsequent T cell differentiation (*Figure 4—figure supplement 1C*), implying that sustained expression of IRF4 in DCs can blunt the effects of cAMP on their function. cDC2s from mice with deletion of IRF4 in CD11c-expressing cells (IRF4$^{\Delta CD11c}$ mice, generated by breeding of *Irf4* floxed mice (*Irf4*$^{fl/fl}$) with CD11c-Cre mice) also induced Th17 differentiation spontaneously (*Figure 4F*), or if treated with CPT or curdlan. However these cells had minimal or no significant increases of IL-1β or IL-6 (*Figure 4G,H*), cytokines that provoke Th17 differentiation (*Chen et al., 2008*).

IRF5 in macrophages has induces Th17 response (*Krausgruber et al., 2011*). We thus tested whether IRF5 in cDC2s affects the pro-Th17 phenotype. CPT treatment of OVA-pulsed cDC2 from *Irf5*$^{-/-}$ mice did not provoke a Th17 response (*Figure 4—figure supplement 2A*). Even though *Irf4* and *Klf4* levels were inhibited, *Irf8* was unchanged and *Crem* was induced (*Figure 4—figure supplement 2B*). Thus, even in the absence of IRF4, IRF5 was required for the cAMP-induced pro-Th17 phenotype. *Irf5* levels were similar in *Irf4*$^{-/-}$ and WT mice, indicating that IRF5 expression is independent of IRF4.

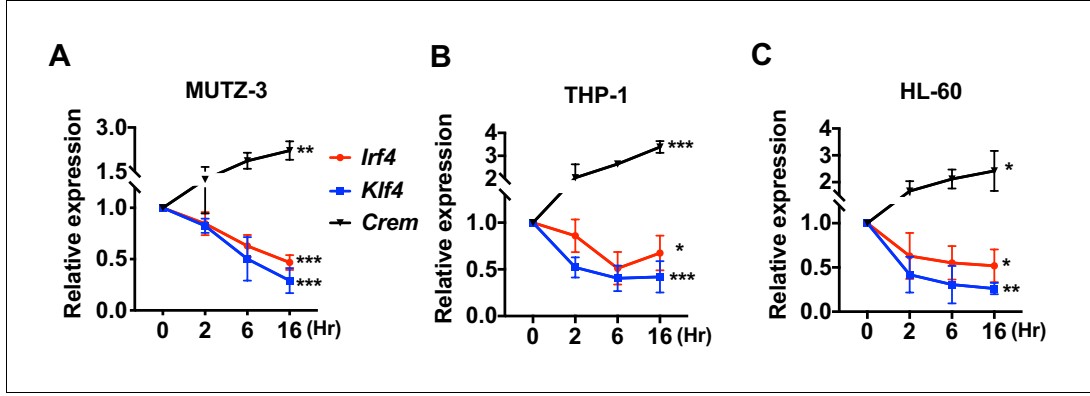

**Figure 2.** Decreased expression of *Irf4* by CPT in human DC-like cells DC-like cells were differentiated from (A) MUTZ-3, (B) THP-1, and (C) HL-60 cell lines and treated with CPT for the indicated time. Relative expression of *Irf4*, *Klf4*, and *Crem* were analyzed. Data are representative of three independent experiments; Two-way ANOVA \*p<0.05, \**p<0.01, \***p<0.001.

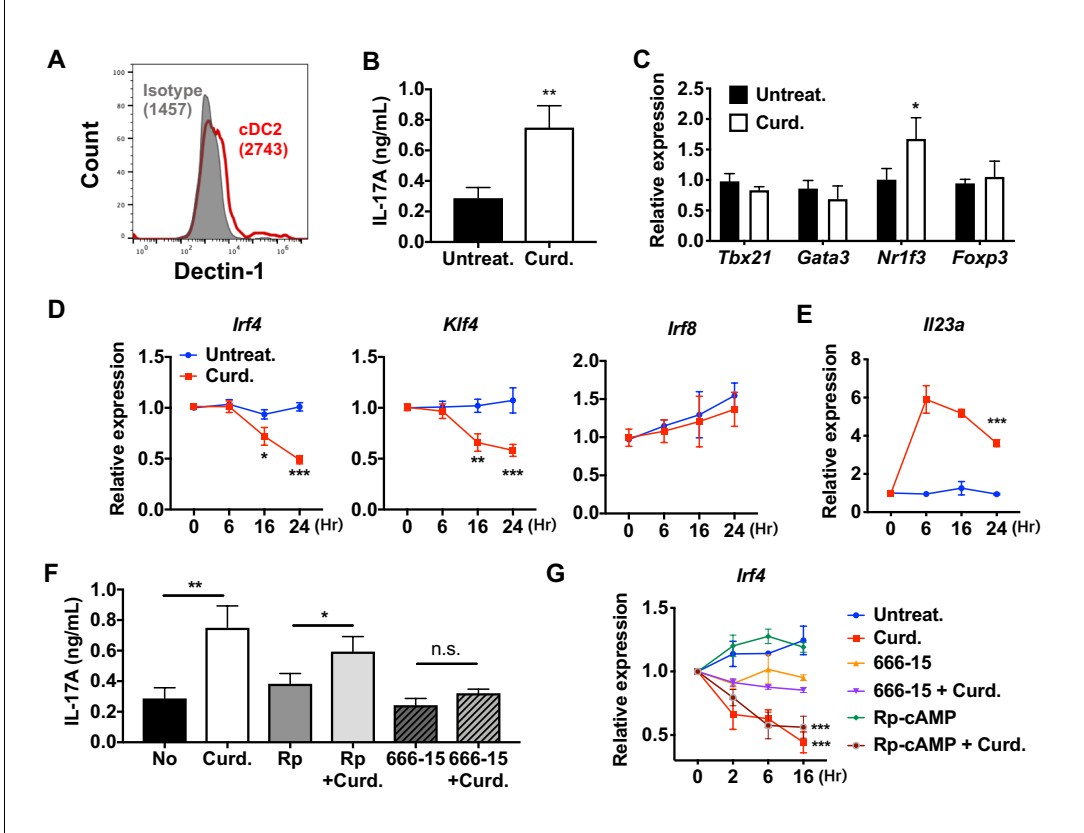

**Figure 3.** Stimulation of DCs via Dectin-1 (a PPR) regulates IRF4 and KLF4 expression, and induce Th17 differentiation. (A) Expression of Dectin-1 on WT cDC2s. Numbers indicate mean fluorescence intensity (GeoMFI). (B) IL-17A levels produced by OT-II cells co-cultured with WT splenic cDC2s treated with or without curdlan (10 μg/ml). (C) qPCR of lineage commitment factors in OT-II cells co-cultured with curdlan-treated and untreated WT cDC2s. qPCR of (D) TFs and (E) IL-23 in the WT cDC2s treated with curdlan for the indicated time points. Two-way ANOVA; n = 3 in each group. (F) IL-17A levels produced by OT-II cells co-cultured with WT BM-APC pre-treated with Rp-cAMP (50 μM), or 666–15 (1 μM) 16 hr prior to curdlan treatment. (G) qPCR of *Irf4* in WT cDC2s treated with or without curdlan in the presence of inhibitors of CREB (666–15, 1 μM) or PKA (Rp-cAMP, 50 μM). Two-way ANOVA; n = 3 in each group. Data are representative of three independent experiments; *p<0.05, **p<0.01, ***p<0.001.

The online version of this article includes the following figure supplement(s) for figure 3:

**Figure supplement 1.** Curdlan does not affect the expression of *Irf4* and subsequent T cell differentiation in cDC1s.

Overall, these results emphasize that the deletion or substantial inhibition of IRF4 levels in DCs is necessary but not sufficient for provoking Th17 differentiation, and that IRF5 expression in DCs is necessary but insufficient for cAMP to provoke Th17 bias.

## IRF4 mediates cAMP-promoted transcriptional events

To assess the mechanisms for cAMP-PKA-mediated suppression of IRF4 expression, we initially performed transcriptional profiling (RNAseq) of untreated or CPT-treated splenic cDC2s from WT mice, global *Irf4* KO mice (*Irf4*[-/-]), and Irf4i mice treated with Dox. Treatment of splenic cDC2s from WT mice with CPT altered expression of 2991 genes (FDR < 0.05, *Figure 5A*), results resembling those previously observed for BM-APCs (*Datta et al., 2010*). Expression of 2454 genes was significantly altered in *Irf4*[-/-] cDC2 s compared to WT cDC2s but expression of only 1401 genes was altered in *Irf4*[-/-] cDC2 s with restored expression of *Irf4* (Dox) (*Figure 5A*; *Supplementary file 1*: containing Supplementary Tables 1-3). Surprisingly, only 24% of the genes altered in the *Irf4*[-/-] cDC2s were restored by re-expression of IRF4 in Dox-treated Irf4i cells, and only 42% of the Dox-dependent genes were altered in the *Irf4*[-/-] cells. These results are not related to the stringency of the multiple testing correction as the discordance remained (30% and 46%, respectively) even without a correction for FDR. The findings are consistent with prior data for genes that display a bimodal pattern of

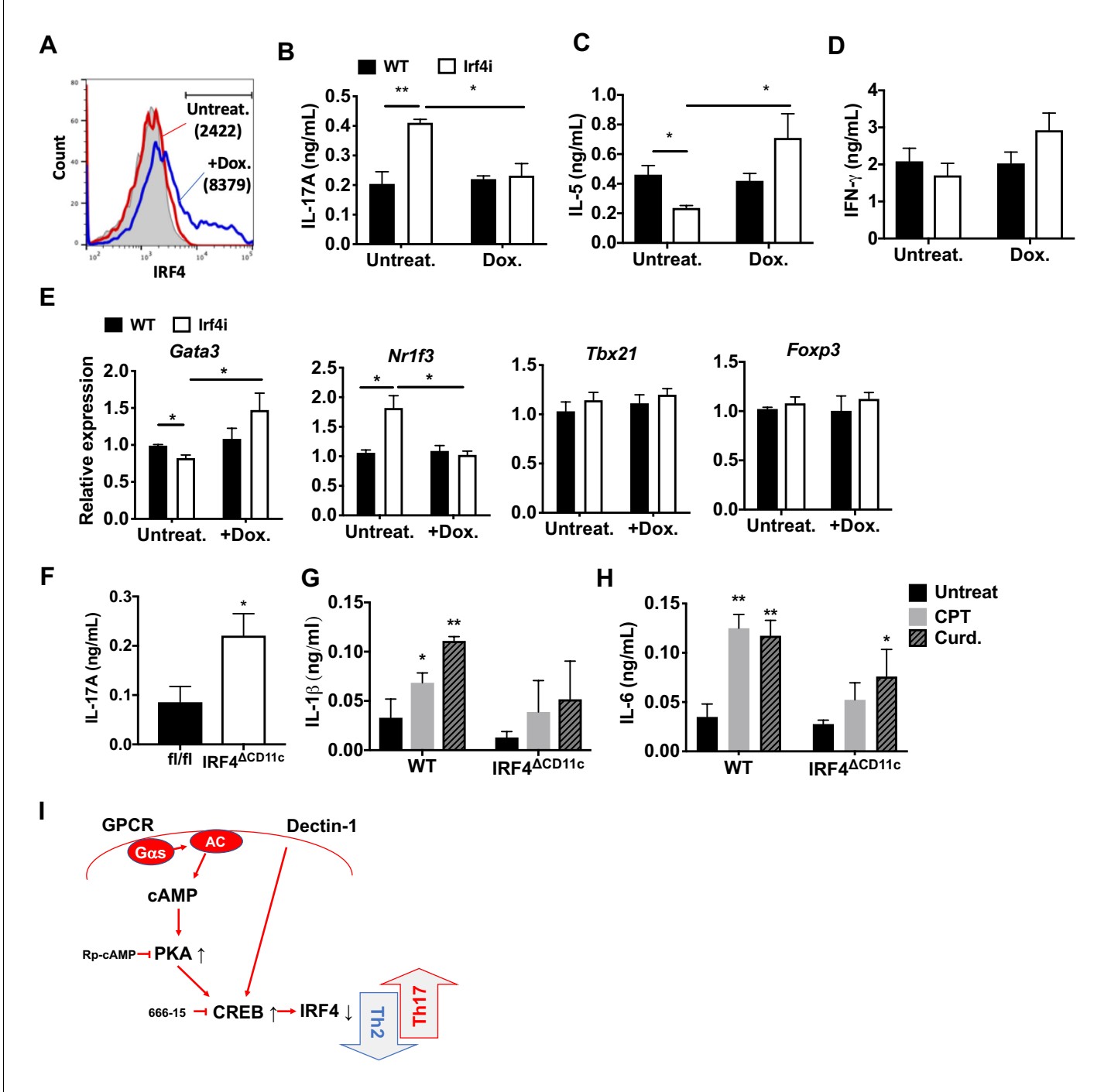

**Figure 4.** Decreased IRF4 expression in cDC2s promotes pro-Th17 phenotype. (**A**) IRF4 expression in the Irf4i cDC2s treated with or without doxycycline (Dox, 200 ng/ml) for 16 hr. (**B–D**) IL-17A, IL-5 and IFN-γ levels and (**E**) T cell lineage commitment factors from the re-stimulated OT-II cells co-cultured with cDC2s from WT and Irf4i mice under the conditions described above. (**F**) IL-17A levels from anti-CD3/28 Ab-stimulated OT-II cells co-cultured with IRF4^ΔCD11c cDC2s. (**G**) IL-1β and (**H**) IL-6 level in WT and Irf^ΔCD11c cDC2s after treatment of CPT or Curdlan for 24 hr. Data are representative of three independent experiments; *p<0.05, **p<0.01. (**I**) A schematic diagram of Th2 inhibition and pro-Th17 induction by PRR-independent and -dependent signals.

The online version of this article includes the following figure supplement(s) for figure 4:

**Figure supplement 1.** Decreased IRF4 expression in BM-APCs promotes pro-Th17 phenotype and sustained IRF4 expression blocks cAMP-induced Th17 bias.

**Figure supplement 2.** Loss of IRF5 in cDC2s inhibited CPT induced pro-Th17 phenotype.

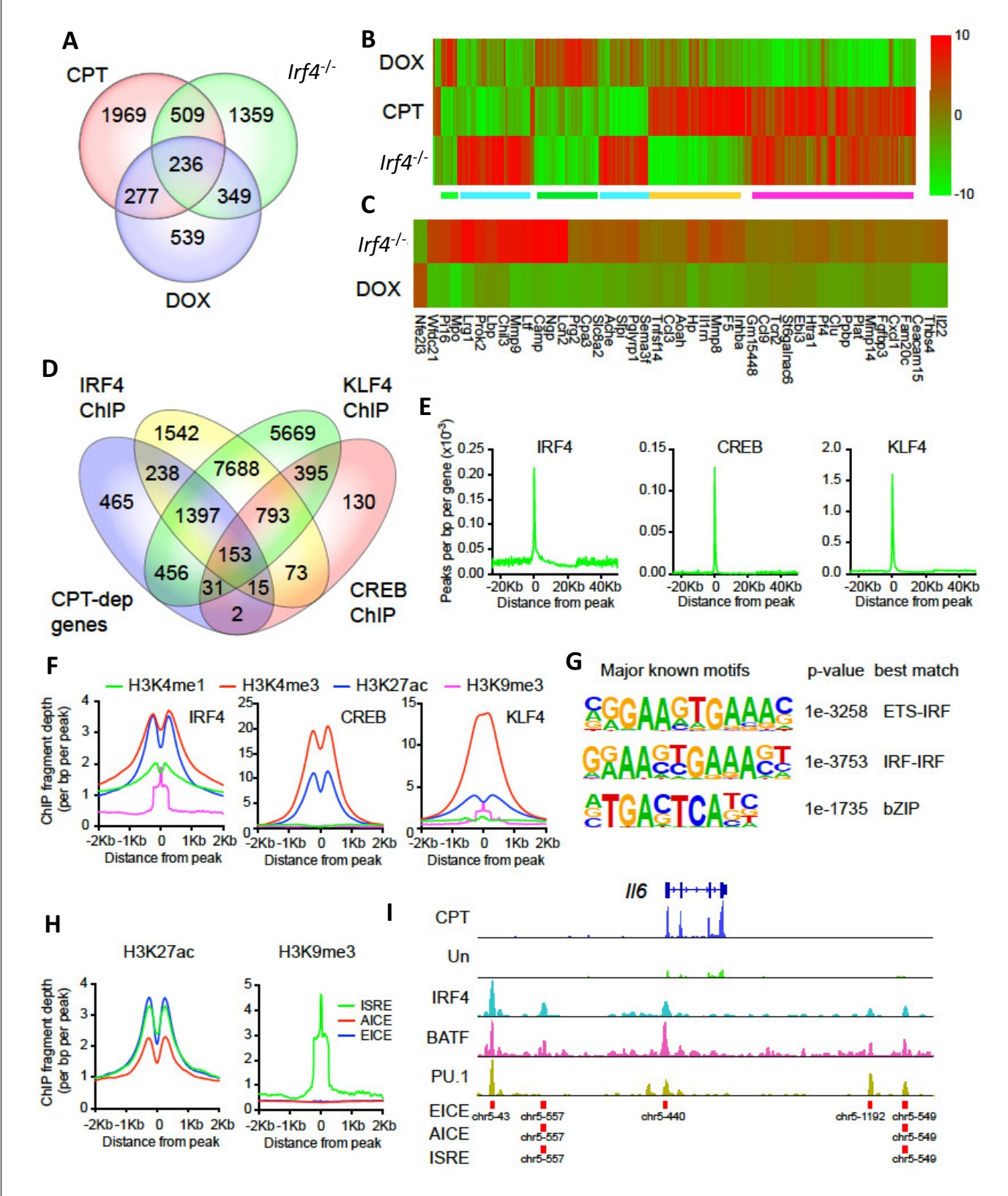

**Figure 5.** CPT and IRF4 transcriptomic effects and analysis of genome-wide binding of IRF4, CREB and KLF4. (**A**) Venn diagram showing overlap of genes altered by CPT treatment (50 μM, 16 hr) of WT splenic cDC2s, with splenic cDC2s derived from *Irf4-/-* mice, and splenic cDC2s from *Irf4-/-* mice

*Figure 5 continued on next page*

*Figure 5 continued*

that have been induced with Dox (200 ng/ml, 16 hr). (B) Heatmap showing expression (log2 fold- change) of the 745 genes common to CPT and *Irf4-/-* cDC2s. Colored bars under the heatmap indicate clusters of genes with similar expression patterns. (C) Heatmap showing expression (log2 fold-change) of secreted genes that are altered in *Irf4-/-* and Dox datasets. (D) Venn diagram showing overlap of CPT-dependent genes with ChIPseq peaks for IRF4, CREB1 and KLF4. (E) Metagene analysis of the localized binding of IRF4, CREB1 and KLF4 to a synthetic gene. The synthetic gene is 25 kB in length and is flanked by 25 kB of upstream and downstream sequence. The transcriptional start site is indicated at 0. The plot shows the number of peaks per bp per gene (x10$^{-3}$). (F) Co-localization of histone epigenetic modifications H3K4me1, H3K4me3, H3K27ac and H3K9me3 at IRF4, CREB and KLF4 peaks. The graphs show the ChIP fragment depth relative to the center of the TF peak. (G) Known transcription factor motifs identified in the IRF4 binding peaks. Height of the letter indicates its conservation. The p-value for the motif and its best match are shown. (H) Co-localization of H3K27ac and HeK9me3 modifications at the peaks with the three IRF4 motifs (ISRE, EICE, AICE). (I) IRF4, BATF, and PU.1 binding to the IRF4-super enhancer at the *Il6* locus. The *Il6* gene structure is shown at the top and RNAseq reads from untreated and CPT-treated cDC2s are shown in green and blue. Locations of individual motifs are indicated below the binding.

The online version of this article includes the following figure supplement(s) for figure 5:

**Figure supplement 1.** Examples of genes from transcriptional clusters.
**Figure supplement 2.** Transcriptional activation/repression prediction analysis.
**Figure supplement 3.** Prevalence and predictive value of IRF4 binding site subpopulations.
**Figure supplement 4.** Expression from, chromatin structure of, and histone modifications at the *Irf4* and *Klf4* loci in cDC2s.

expression as a function of IRF4 concentration in B and T cells (*Iwata et al., 2017*; *Krishnamoorthy et al., 2017*; *Ochiai et al., 2013*).

Transcriptional network analysis showed that the predominant TF driving CPT-dependent genes was Creb1, which appeared to regulate 1134 genes (38%, z-score 236) (*Supplementary file 1*: containing Supplementary Table 4). Unexpectedly, CREB1 was also the major TF 'driver' of genes altered in *Irf4-/-* cells and Dox-treated Irf4i cells (592 and 442 such genes, 24% and 32%, z-scores 233 and 240, respectively) (*Supplementary file 1*: containing Supplementary Tables 5-6). These observations implicate CREB1 as a cofactor for IFR4-regulated genes and suggest a link between cAMP signaling and IRF4-regulated transcription in DCs.

The genes altered by CPT treatment or loss of IRF4 significantly overlapped (p<0.0001 by Chi-square): 745 genes (*Supplementary file 1*: containing Supplementary Table 7) were in common (*Figure 5A*), a greater number than between the *Irf4-/-* and Dox-treated cells (585 genes, *Supplementary file 1*: containing Supplementary Table 8). Clustering analysis revealed distinct classes of changes among the 745 common genes (*Figure 5B*). A large cluster of genes was concordantly induced (*Figure 5B* magenta) and a smaller cluster of genes was reduced (*Figure 5B* green) by CPT or loss of IRF4, with opposite responses by IRF4 re-expression. These two clusters may indicate genes indirectly regulated by cAMP via its suppression of IRF4 expression. Examples of genes in these clusters are the IL-1 receptor antagonist (*Il1rn*) and C-C motif chemokine receptor 2 (*Ccr2*) (*Figure 5—figure supplement 1A,B*). Other clusters are discordant. that is, decreased by CPT but increased in the *Irf4-/-* cells or vice versa (*Figure 5B*, green and orange). Examples of genes in these clusters are plasminogen activator, urokinase receptor (*Plaur*) and matrix metalloprotease 27 (*Mmp27*) (*Figure 5—figure supplement 1C,D*).

Since CPT treatment, loss of IRF4 or restoration by Dox treatment all changed the ability of cDC2s to promote Th bias, we reasoned that key mediators of this bias would be among the 236 genes that were common to the three datasets (*Supplementary file 1*: containing Supplementary Table 9). Transcriptional network analysis and enrichment analysis for pathways and processes of these 236 common genes revealed that the most significantly enriched TF was CREB1 (*Supplementary file 1*: containing Supplementary Table 10) and the most significantly enriched pathways related to cell adhesion, CCL2 signaling, immune cell migration, myeloid differentiation, neutrophil activation, and inter-cellular interactions in COPD (*Supplementary file 1*: containing Supplementary Table 11). Network enrichment of processes centered on angiogenesis and immune cell activation/response pathways (*Supplementary file 1*: containing Supplementary Table 11). The two highest enriched GO processes were secretion-related (p<10$^{-25}$), suggesting involvement of secreted mediators of Th differentiation (*Supplementary file 1*: containing Supplementary Table 11). We therefore intersected this group of genes with curated and highly-likely predicted secreted

proteins from the MetazSecKB database (*Meinken et al., 2015*) and found that secreted proteins were highly enriched (59 of the 236 common genes (25%), p<0.0001). We subdivided these genes into those with concordant or discordant regulation between CPT treatment and *Irf4*⁻/⁻ (*Supplementary file 1*: containing Supplementary Table 12). Of the 22 concordant genes among the data sets, all were up-regulated by treatment with CPT or in *Irf4*⁻/⁻, suggesting that promotion of Th17 differentiation requires production of a Th17 mediator rather than loss of a Th2 mediator. Since the loss of *Irf4* can lead to Th17 differentiation, we inspected the secreted genes for ones induced by IRF4 loss and restored by IRF4 'rescue' (*Figure 5C*). We also intersected the differentially expressed genes with predicted plasma membrane proteins and found that these are significantly enriched (82 of the 236 common genes (34%), p<0.0001) (*Supplementary file 1*: containing Supplementary Table 13). These results imply that many of the cAMP-dependent transcriptional changes may be a consequence of the repression of IRF4 expression. Furthermore, the IRF4- and cAMP-dependent genes are enriched in secreted and plasma membrane proteins that could modulate T cell differentiation.

## Transcriptional changes correlate with transcription factor binding but not with alteration in chromatin accessibility

To gain a better understanding of the genomic events involved in these transcriptional changes, we performed a global ATACseq analysis for changes in chromatin accessibility (*Supplementary file 1*: containing Supplementary Table 14 and *Figure 5—figure supplement 1*). We observed no significant differences (after multiple-testing correction [FDR < 0.05]) in ATACseq peaks in CPT-treated WT cDC2. *Irf4*⁻/⁻ cDC2 had only one significant ATACseq peak (in the *Rasl11a* gene); re-expression of IRF4 caused changes in 9 ATACseq regions. These results suggest that cDC2 plasticity is not accompanied by gross changes in chromatin structure.

We then analyzed ChIPseq datasets derived from IRF4, CREB, KLF4, and histone modifications in BMDC, cDC2 and related cells (*Supplementary file 1*: containing Supplementary Table 15). We examined the co-localization of binding sites for IRF4, CREB and KLF4 within CPT-regulated genes (*Figure 5D*). Surprisingly, only 7% of CPT-regulated genes (n = 201) contained CREB binding sites, in contrast to 65% with an IRF4 binding site, 74% with a KLF4 binding site, and 56% with both IRF4 and KLF4 binding sites (*Figure 5D*). Combining the expression data with the ChIP data showed that the presence of neither a CREB nor a KLF4 binding site was predictive of up- or down-regulation of expression by CPT treatment) (*Figure 5—figure supplement 2A*). By contrast, the presence of an IRF4 binding site was predictive of both up- and down- regulation by CPT (p=$3.6\times10^{-5}$ and $2.6\times10^{-6}$, respectively), implying that cAMP regulation of genes in cDC2 may be mediated by alterations in IRF4 expression. Indeed, 890 of the 2991 CPT-regulated genes (30%) were also regulated by IRF4 loss or over-expression (*Figure 5A*). A similar analysis with genes altered in the *Irf4*⁻/⁻ and Irf4i cells revealed that IRF4-regulated genes in those cDC2s were not predicted by the presence of a CREB binding site (*Figure 5—figure supplement 2B,C*), but that up-regulated genes in the *Irf4*⁻/⁻ cells and down-regulated genes in the Irf4i cells were highly predicted by IRF4 binding sites (p=$6.9\times10^{-13}$ and p=$1.7\times10^{-14}$, respectively). IRF4 binding sites were less predictive for genes that were decreased in *Irf4*⁻/⁻ cells (p=0.0012) but were predictive for genes induced by IRF4 over-expression (p=$7\times10^{-6}$). The presence of KLF4 binding sites predicted genes that decreased with IRF4 over-expression (p=$7.7\times10^{-5}$).

To further understand the transcriptional regulation, we analyzed the location of the IRF4, CREB and KLF4 binding sites relative to the transcriptional start sites (TSS) and the gene bodies, as well as 25 Kb upstream and downstream of the genes. Both CREB and KLF4 primarily bound to the TSS with very little binding to the gene body, upstream or downstream regions. In contrast, IRF4 showed weaker binding to the TSS but more substantial binding upstream and downstream of the genes (*Figure 5E*). Consistent with this observation, the IRF4, CREB and KLF4 binding sites were associated with H3K4me3 and H3K27ac marks of transcriptionally active promoters (*Jenuwein and Allis, 2001*) but only IRF4 binding sites were associated with the H3K4me1 modification, which is indicative of enhancers, and with the repressive H3K9me3 modification in the middle of the IRF4 peak (*Figure 5F*).

We investigated the known TF motifs bound by IRF4 and found that they fell into three categories: 50% of the sites had ETS:IRF motifs similar to proposed EICEs (*Brass et al., 1996*; *Ochiai et al., 2013*), 24% had duplicated IRF motifs similar to ISREs (*Bovolenta et al., 1994*; *Ochiai et al., 2013*),

and 23% had bZIP motifs similar to AICEs (*Figure 5G* and *Figure 5—figure supplement 3A*). The sub-populations of IRF4 binding sites had similar association with H3K4me1 and H3K27ac enhancer marks but different co-localizations with histone repressive marks, in that only IRF4 at ISRE sites was associated with the transcriptionally repressive H3K9me3 mark (*Figure 5H*). IRF4 may thus regulate gene expression in a context-specific manner as well as by differential binding dynamics dependent on its interactions with its binding partners (*Krishnamoorthy et al., 2017*; *Ochiai et al., 2013*).

Given this difference in association with transcriptionally repressive histone modifications, we performed a similar analysis with transcriptional changes in response to treatment with CPT (*Figure 5—figure supplement 3B*). The presence of an AICE most strongly predicted up-regulation by CPT treatment (p=$1.7\times10^{-6}$), whereas all three response elements predicted up-regulation in *Irf4*$^{-/-}$ cells, and down-regulation in Dox-treated cells. Thus, all three sites may mediate repression by IRF4. The AICE and ISRE, but not the EICE, also predicted gene induction in the Irf4i cells, suggesting that these two elements can confer gene induction by IRF4 (*Figure 5—figure supplement 3B*). Many of the IRF4 sites clustered together. Using the Young lab criteria, we found evidence for 106 super enhancers with multiple IRF4 binding sites (*Supplementary file 1*: containing Supplementary Table 16) (*Whyte et al., 2013*). Inspection of these super enhancers revealed that one was located in the *Il6* locus and had multiple binding sites for IRF4, BATF and PU.1; these peaks corresponded to EICEs, AICEs and ISREs identified in the motif analysis (*Figure 5I*). Localization of IRF4, CREB, KLF4, H3K4me3 and H3K27ac peaks and the presence of open chromatin (measured by ATACseq) in the vicinity of the *Irf4* and *Klf4* genes indicated that the *Klf4* gene, but not the *Irf4* gene, may be a direct target for CREB (*Figure 5—figure supplement 4*).

These genomic data were thus consistent with a role for IRF4 binding to and regulation of cAMP- and IRF4-dependent genes. The results also suggested that CREB does not directly regulate many of these cAMP-dependent genes. The data also implied that IRF4-dependent transcriptional events are modulated by the partner TF as IRF4 homodimers or heterodimers with ETS or bZIP proteins predicted to contribute to gene repression, but only IRF4 homodimers and bZIP heterodimers mediating gene induction.

### Adoptive transfer of HDM-pulsed, CPT- or curdlan-treated cDC2s induce a Th17 bias and neutrophilic infiltration

We hypothesized that activation of DCs via the cAMP pathway would modify their Th-inducing properties in vivo. To test this hypothesis, we evaluated the impact of CPT on DCs in the induction of Th17 bias by using DC-based adoptive transfer (*Lambrecht and Hammad, 2012*) (*Figure 6A*). HDM- and curdlan-treated cDC2s served as negative and positive controls, respectively, of Th17 induction. Intranasal (i.n.) transfer of HDM-pulsed, CPT-treated or curdlan-treated WT cDC2s followed by HDM i.n challenge increased the production of IL-17A in the lung and airway of WT recipient mice (*Figure 6B,C*). However, we did not observe differences in levels of IL-4 (>0.04 ng/ml) or IFN-γ (>0.02 ng/ml) in the various groups. We detected an increased number of neutrophils in the bronchoalveolar lavage (BAL) fluid (*Figure 6D–G*) and expression of *Cxcl2* and *Cxcl3*, whose expression accompanied neutrophilic infiltration in the CPT- or curdlan-treated cDC2 groups (*Figure 6H*). Expression of eosinophilic infiltration-related genes, *Cd24a* and *Dpp4*, was decreased in those groups (*Figure 6H*). CPT- or curdlan-treated WT cDC2s transferred to WT recipient mice also had increased airway resistance (*Figure 6I*). CPT or curdlan treatment of OVA-pulsed cDC2s ex-vivo prior to their i.n transfer yielded a pattern similar to that observed for HDM (*Figure 6—figure supplement 1*). Induction of Th17 in the lungs by CPT or curdlan with two different antigen systems thus yielded a shift from a Th2 toward Th17 pulmonary response with a mixed eosinophil/neutrophil inflammatory infiltrate.

## Discussion

The dominant role of PRRs has been the paradigm for the activation and maturation of DCs with resultant antigen processing and presentation, up-regulation of co-stimulatory molecules, and secretion of cytokines and other immunomodulatory factors; such factors contribute to the differentiation of naïve T cells to effector subpopulations, such as CTL or Th subsets (*Akira et al., 2006*; *Merad et al., 2013*). Our results indicate that cAMP, via a PRR-independent pathway, and the microbial product, curdlan, via a PRR-dependent pathway, converge on CREB in cDC2s but not cDC1s.

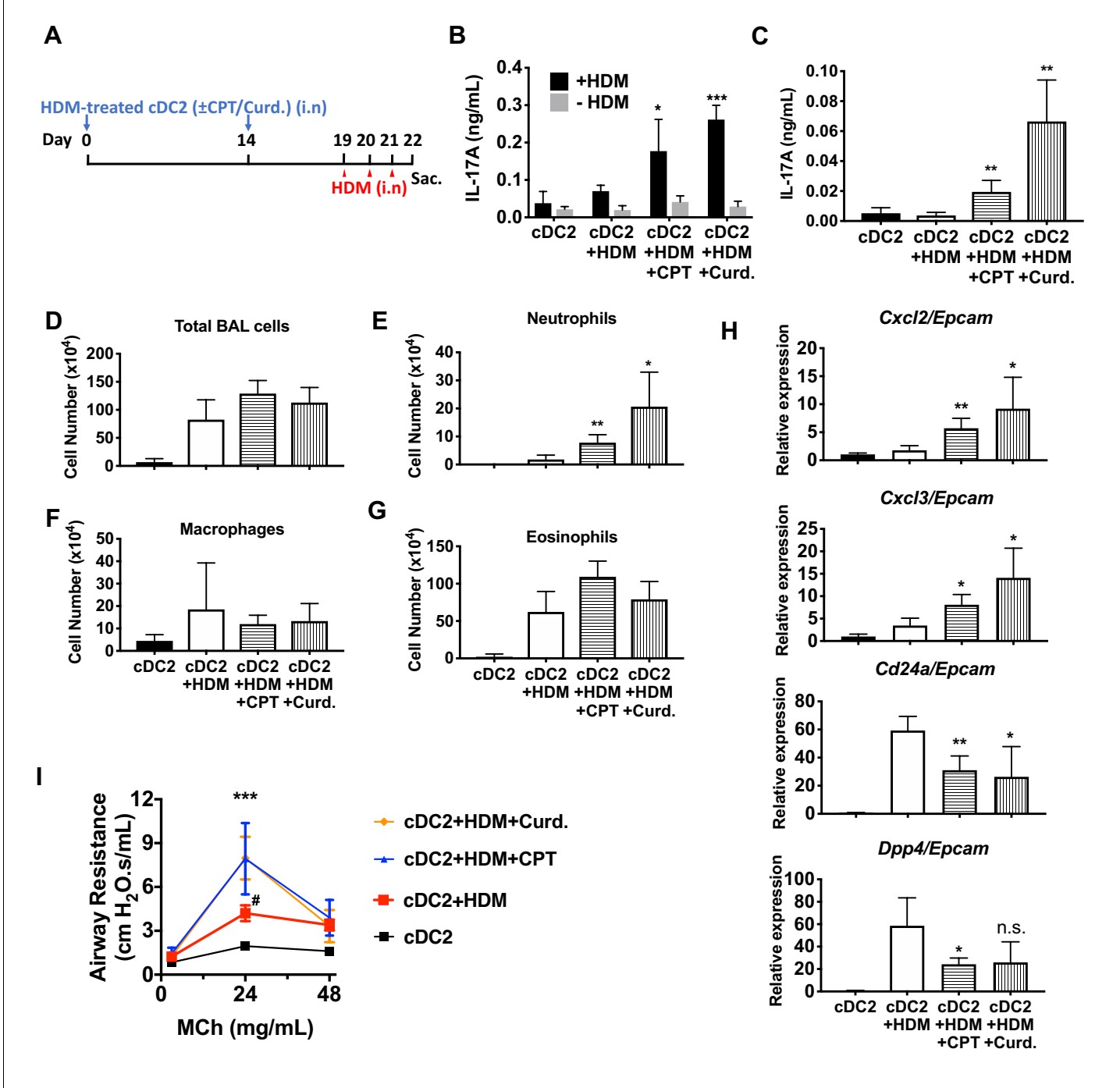

**Figure 6.** Adoptive transfer of HDM-pulsed, CPT or curdlan-treated cDC2 induces a Th17 bias and neutrophil infiltration in WT recipients. (A) Schematic of adoptive transfer protocol. WT cDC2s were incubated with HDM (50 µg/ml) in the presence of CPT or curdlan prior to i.n. transfer to WT mice (B6 mice, $1 \times 10^6$ cells/recipient) on day 0 and 14. HDM (12.5 µg/mouse) was used for the i.n. challenge on days 19, 20 and 21. On day 22, lungs from each group were harvested and processed a single cell suspension. (B) IL-17A level in the HDM (50 µg/ml)-stimulated lung cells. BAL fluid was analyzed for (C) IL-17 level (ELISA). (D) Total cells (E) neutrophils, (F) macrophages and (G) eosinophils number were counted in the BAL fluid. (H) Relative expression of Neut (*Cxcl2* and *Cxcl3*)- and/Eos (*Cd24a* and *Dpp4*)- infiltration related genes in the lung tissue. Expression of each gene was normalized by expression of epithelial specific housekeeping gene, *Epcam*. (I) Airway resistance after MCh challenge was measured in the various experimental groups. Data were collected from four animals in each group and are representative of two independent experiments; *p<0.05, **p<0.01, ***p<0.001 by two-tailed Student's t-test.

The online version of this article includes the following figure supplement(s) for figure 6:

**Figure supplement 1.** Adoptive transfer of OVA-pulsed, CPT or curdlan-treated cDC2s induces a Th17 bias and neutrophilic asthma in WT recipients.

Consequently, *Irf4* and *Klf4* levels are repressed, resulting in a pro-Th17 phenotype of cDC2s (i.e., cDC17s). These results define a novel molecular event by which cDC2s undergo re-programming and hence affect Th cell differentiation, that is, the induction of Th2 vs. Th17 bias (*Figures 1* and *3* and *Figure 1—figure supplement 4*).

Prior emphasis on the role of IRF4 in DCs have mainly focused on the up-regulation of IRF4 expression. In the current studies, we discovered that IRF4 expression is repressed by cAMP or curdlan signaling. DCs signaled in this manner skew the consequent T cell responses towards the Th17 effector lineage. In contrast, when DCs signal through FcγRIII, IRF4 expression is up-regulated and leads those DCs to skew T cell responses towards a Th2 effector lineage (*Williams et al., 2013*). The current results thus identify cAMP-PKA-CREB signaling as a non-PRR pathway that reprograms DCs toward Th17, results that complement evidence that reduced cAMP signaling in DCs provokes Th2 differentiation (*Lee et al., 2015*). Furthermore, increasing the cAMP concentration in pro-Th2 cDC2s can switch them toward the cDC17 phenotype. DCs thus appear to integrate stimuli from their extracellular environment to control their expression of IRF4, which in turn, regulates the ensuing T cell response.

The effects of cAMP were observed in two different types of DCs; splenic cDC2s and BM-APCs but not in cDC1s cells (*Figure 1* and *Figure 1—figure supplements 1* and *4*) and are achieved by the inhibition of *Irf4* and *Klf4*, TFs that are prerequisites for the pro-Th2 DC phenotype of cDC2 (*Gao et al., 2013*; *Tussiwand et al., 2015*; *Williams et al., 2013*). cDC2s from Gnas$^{\Delta CD11c}$ mice (cells with prominently decreased cAMP signaling) have higher levels of *Irf4* and *Klf4* than do cDC2s from WT mice (*Figure 1—figure supplement 4*). The *Irf4* and *Klf4* transcript and IRF4 protein levels were inhibited by multiple agonists that increase cAMP and by curdlan (cAMP-independent) activation of the PRR Dectin-1 in WT cDC2s, results that suggest convergent signaling to inhibit *Irf4* and *Klf4* for the cDC17 phenotype (*Figure 3*). Based on previous data (*Bedoui and Heath, 2015*; *Persson et al., 2013*; *Schlitzer et al., 2013*), we were surprised that BM-APCs and cDC2s from two mouse strains in which IRF4 had been genetically-deleted could provoke a strong Th17 response in the presence of very low IL-1 and IL-6 secretion, suggesting that this pathway of Th17 induction is independent of these cytokines (*Figure 4F–H*). Curdlan provoked neither cAMP synthesis nor PKA phosphorylation, but activates CREB (*Elcombe et al., 2013*), possibly via MAPK signaling (*Kim et al., 2016*). We confirmed by using adoptive transfer that pro-Th17 DCs induction by cAMP-independent or PRR-dependent signaling pathways resulted in increased neutrophils and IL-17A production in the airway (*Figure 6*, *Figure 6—figure supplement 1*).

Our results thus indicate that: 1) inhibition of the pro-Th2 DC cDC2 phenotype by cAMP agonists promotes induction of the pro-Th17 DC cDC17 phenotype, and 2) reduction of IRF4 levels is necessary but not sufficient for induction of the pro-Th17 transcriptional program in DCs by both the PRR-dependent and cAMP-dependent signaling pathways and by cAMP, also requires IRF5. We propose that cDC2 cells appear to include two subpopulations: cDC17 (CD11c$^+$, CD11b$^+$, IRF4$^-$ or IRF4$^{low}$, IRF5$^+$), and cDC2 (CD11c$^+$, CD11b$^+$, IRF4$^+$, KLF4$^+$).

Further, our data indicate that endogenous IRF4 is a transcriptional repressor of a subset of cAMP-dependent genes in cDC2s, based on the following: 1) its loss causes up-regulation of cAMP-dependent gene expression, 2) presence of an IRF4 binding site is highly predictive of a gene that is regulated by cAMP, and 3) an IRF4 binding site predicts that a gene increases in *Irf4*$^{-/-}$ cDC2s and decreases in IRF4 restored cells. Overexpression of IRF4 restored the repression of these genes but also induced expression of genes not normally regulated by IRF4. The presence of a KLF4 binding site in the context of IRF4 over-expression predicts down-regulation of gene expression, consistent with the known role of KLF4 as a transcriptional repressor. We found that IRF4 binding sites tended to be associated with enhancers as well as promoters and have three known IRF4 binding motifs (50% EICE, 23% AICE, 24% ISRE). While all three sites were associated with transcriptional activity, only the ISRE was associated with transcriptional repression. The AICE was most strongly associated with up-regulated genes by CPT, perhaps indicating a role for the bZIP binding partner BATF/JUN in this response. We therefore conclude that IRF4 is a major repressor of cAMP-stimulated gene expression in cDCs.

Thus, a portion of cAMP-stimulated genes in cDCs appear to be indirectly regulated via suppression of *Irf4* expression. The mechanism involved in the repression of *Irf4* is unknown. Intriguingly, the orphan nuclear receptor *Nr4a* represses *Irf4* expression in CD8$^+$ T cells by binding to multiple sites in the *Irf4* promoter (*Nowyhed et al., 2015*). OVA-challenged *Nr4a1* KO mice have increased airway

inflammation, elevated Th2 cytokines and eosinophils in BAL fluid (*Kurakula et al., 2015*). In Leydig cells, cAMP activates expression of *Nr4a1* (*Mori Sequeiros Garcia et al., 2015*) but we find that CPT treatment represses *Nr4a1* expression in cDC2s so this is unlikely to be the mechanism. In contrast, the related family member *Nr4a3* stimulates *Irf4* expression in BMDC (*Nagaoka et al., 2017*), so NR4A family members may regulate *Irf4* expression through differential recruitment of co-repressor and co-activators. Certain other transcriptional regulators (e.g., NF-κB, NFAT, SP1 and STAT4) activate *Irf4* expression (*Bat-Erdene et al., 2016*; *Boddicker et al., 2015*; *Lehtonen et al., 2005*; *Sharma et al., 2002*). How cAMP signaling interacts with these pathways in cDC2s is not known but VIP/cAMP signaling interferes with IL-12/JAK2/STAT4 and NF-κB signaling in T cells and macrophages (*Leceta et al., 2000*; *Liu et al., 2007*). The regulation could also be epigenetic: expression of *Nr4a1* and *Irf4* are repressed by HDAC7 in naïve T cells (*Myers et al., 2017*).

Reduction of IRF4 expression by cAMP elevation or genetic knockout induced a number of genes encoding secreted proteins in cDC2s, some of which have roles in Th differentiation and immune regulation. For example, the cAMP- and IRF4-dependent genes Lipocalin-2 (*Lcn2*) (*Hau et al., 2016*) and Leucine-rich alpha two glycoprotein (*Lrg1*) (*Urushima et al., 2017*) may have functional roles in Th17 differentiation (*Figure 5C*), perhaps contributing to the independence of Th17 induction from IL-1 and IL-6 (*Figure 4F–H*). Similarly, the *Ebi3* (*Wang et al., 2018*; *Böhme et al., 2016*; *Chung et al., 2013*), *Il1rn* (*Ikeda et al., 2014*; *Rogier et al., 2017*), *Mmp14* (*Baranov et al., 2014*; *Gawden-Bone et al., 2010*), and *Pf4* genes (*Gerdes et al., 2011*; *Xia and Kao, 2003*) have been implicated in Th17 function, and the *Thbs4* gene is associated with altered lung function (*Muppala et al., 2015*; *Panasevich et al., 2013*).

Microbial stimulation most likely does not underlie the cAMP-mediated Th2 and Th17 responses for several reasons: 1) the in vivo Th2 and Th17 differentiation were generated in vitro in the presence of antibiotics, 2) the Gnas$^{\Delta CD11c}$ mice were co-housed with Gnas$^{fl/fl}$ littermates, which showed no Th bias, 3) Gnas$^{fl/fl}$ BM-APCs and cDC2s display a WT phenotype under the same conditions and 4) WT and Gnas$^{\Delta CD11c}$ BM-APCs and cDC2 could be reprogrammed in vitro to a pro-Th17 phenotype by multiple cAMP-elevating agents (in the presence of antibiotics).

As part of homeostatic regulation, cAMP is regulated by GPCRs that recognize host-derived ligands, such as neurotransmitters, hormones, chemokines, metabolites (e.g., fatty acids, lipid mediators, nucleosides/nucleotides *Marinissen and Gutkind, 2001*), complement cleavage fragments, cholera (*Datta et al., 2010*), pertussis and heat-labile toxins (*Bharati and Ganguly, 2011*), and bacterial formylated peptides (*Cattaneo et al., 2013*). Thus, GPCRs that alter cellular cAMP concentrations likely have a role in innate immunity and Th differentiation. However, the current studies do not reveal the physiologically most important GPCR(s) that mediate these responses.

GPCR pathway drugs (*Sriram and Insel, 2018*) and others that regulate cAMP levels, including cyclic nucleotide phosphodiesterase (PDE) inhibitors, may alter Th2/Th17 bias and potentially may cause immunological side effects, or alter diseases that involve Th2 or Th17 responses. Drugs that increase cAMP levels in DCs perhaps facilitate recovery from bacterial infections such as *K. pneumonia* and *P. aeruginosa,* and fungal infections, for example, *Candida albicans* (*Kumar et al., 2013*), for which Th17 response is beneficial. On the other hand, drugs that decrease cAMP levels in DCs and therefore enhance Th2 immunity, might facilitate treatment of helminth infections. The inhibition of IRF4 by agonists that increase intracellular cAMP may also have implications for other disease settings. IRF4 is a key regulator at multiple steps in B-cell differentiation and development (*Klein et al., 2006*; *Ochiai et al., 2013*; *Sciammas et al., 2006*) and is an oncogene in multiple myeloma (*Shaffer et al., 2009*), to a lesser degree in Hodgkin and non-Hodgkin lymphomas as well as in chronic lymphocytic leukemia (*Boddicker et al., 2015*; *Shaffer et al., 2008*; *Shukla et al., 2013*). The current findings suggest that new (or perhaps repurposed) agonists of Gαs-linked GPCR drugs might be beneficial for the inhibition of IRF4 in B-cell derived malignancies.

In summary, the current studies revealed that both cAMP-PKA-CREB and PRR-CREB signaling reprogram DCs by regulating a set of TFs that control cDC2 and cDC17 phenotypes and subsequent Th2 or Th17 bias. These findings thus identify the cAMP pathway as a previously unappreciated PRR-independent, transcriptional mechanism for Th bias by DCs and suggest new therapeutic approaches to control immune responses.

# Materials and methods

**Key resources table**

| Reagent type (species) or resource | Designation | Source or reference | Identifiers | Additional information |
|---|---|---|---|---|
| Strain, strain background (*Mus musculus*) | C57BL/6J (B6) | The Jackson Laboratory | RRID:IMSR_JAX:000664 | |
| Strain, strain background (*Mus musculus*, B6) | CD11c-Cre | The Jackson Laboratory | RRID:IMSR_JAX:008068 | |
| Strain, strain background (*Mus musculus*, B6) | OT-II | The Jackson Laboratory | RRID:IMSR_JAX:004194) | |
| Strain, strain background (*Mus musculus*, B6) | IL17A-eGFP | The Jackson Laboratory | RRID:IMSR_JAX:018472 | |
| Strain, strain background (*Mus musculus*, B6) | *Irf4*$^{fl/fl}$ | The Jackson Laboratory | RRID:IMSR_JAX:009380 | |
| Strain, strain background (*Mus musculus*, B6) | *Gnas*$^{fl/fl}$ | Lee Weinstein (NIH) | MGI:3609165 PMID: 15883378 | |
| Strain, strain background (*Mus musculus*, B6) | *4Get* | Dr. M. Kronenberg (LAI, San Diego, CA) | RRID:IMSR_JAX:004190 | |
| Strain, strain background (*Mus musculus*, B6) | *Irf4*-inducible (*Irf4*$^{-/-}$) (Irf4i) | Dr. R. Sciammas (University of California, Davis) | PMID: 23684984 | |
| Strain, strain background (*Mus musculus*, B6) | *Irf5*$^{-/-}$ | Dr. I. R. Rifkin (Boston University) | PMID: 25595782 | |
| Strain, strain background (*Mus musculus*, B6) | *Il17a*$^{-/-}$ | Dr. Y. Iwakura (The University of Tokyo, Japan) | MGI:2388010 PMID: 12354389 | |
| Cell line (*Homo-sapiens*) | MUTZ-3 | Dr. Martin L. Yarmush (Rutgers University, New Jersey) | PMID: 29682279 | |
| Cell line (*Homo-sapiens*) | THP-1 | ATCC | RRID:CVCL_0006 | |
| Cell line (*Homo-sapiens*) | HL-60 | ATCC | RRID:CVCL_0002 | |
| Antibody | PE anti-mouse monoclonal CD11c | eBioscience | Cat# 12-0114-82 | FACS (1:200) |
| Antibody | FITC anti-mouse monoclonal CD11c | BioLegend | Cat# 117306 | FACS (1:200) |
| Antibody | APC anti-mouse monoclonal CD8α | BD Bioscience | Cat# 553035 | FACS (1:200) |
| Antibody | FITC anti-mouse monoclonal CD8α | BD Bioscience | Cat# 553031 | FACS (1:200) |

*Continued on next page*

*Continued*

| Reagent type (species) or resource | Designation | Source or reference | Identifiers | Additional information |
|---|---|---|---|---|
| Antibody | APC anti-mouse monoclonal CD11b | eBioscience | Cat# 17-0112-82 | FACS (1:200) |
| Antibody | PE/Cy7 anti-mouse monoclonal CD11b | BioLegend | Cat# 101216 | FACS (1:100) |
| Antibody | PerCP-eFluor 710 anti-mouse monoclonal CD135 | eBioscience | Cat#46-1351-80 | FACS (1:200) |
| Antibody | PerCP-eFluor710 anti-human/mouse monoclonal IRF4 | eBioscience | Cat# 46-9858-80 | FACS (1:200) |
| Antibody | PE anti-human/ mouse monoclonal IRF8 | R and D Systems | Cat# IC8447P | FACS (1:200) |
| Antibody | PE anti-mouse monoclonal IRF5 | Invitrogen | Cat# PCS-201–012 | FACS (1:200) |
| Antibody | PerCP anti-mouse monoclonal CD4 | BD Bioscience | Cat# 553052 | FACS (1:200) |
| Antibody | FITC anti-mouse monoclonal CD4 | BD Bioscience | Cat# 553729 | FACS (1:200) |
| Antibody | PE anti-mouse monoclonal IL-33Rα | BioLegend | Cat# 145303 | FACS (1:200) |
| Antibody | PEcy7 anti-mouse monoclonal CD44 | eBioscience | Cat# 25-0441-82 | FACS (1:200) |
| Antibody | APC anti-mouse monoclonal CD62L | eBioscience | Cat# 17-0621-83 | FACS (1:200) |
| Antibody | APC anti-mouse monoclonal FOXP3 | eBioscience | Cat# 17-5773-82 | FACS (1:200) |
| Antibody | PE anti-mouse monoclonal Dectin-1 | BioLegend | Cat# 144303 | FACS (1:200) |
| Antibody | anti-mouse monoclonal CD3e (2C11) | BioXcell | Cat# BE0001-1FAB | 10 µg/ml |
| Antibody | anti-mouse monoclonal CD28 | BioXcell | Cat# BE0015-5 | 1 µg/ml |
| Commercial assay or kit | EasySep Mouse Naïve CD4+ T Cell Isolation Kit | StemCell Technologies | Cat# 19765 | |
| Peptide, recombinant protein | Recombinant Mouse GM-CSF | BioLegend | Cat# 576302 | 10 ng/ml |
| Peptide, recombinant protein | Recombinant Human GM-CSF | BioLegend | Cat# 572903 | 100 ng/ml |
| Peptide, recombinant protein | TGF-β | R and D systems | Cat# 240-B-002 | 10 ng/ml |
| Peptide, recombinant protein | TNF-α | R and D systems | Cat# 210-TA-005 | 2.5 ng/ml |
| Peptide, recombinant protein | rhIL-4 | BioLegend | Cat# 574002 | 100 ng/ml |

*Continued on next page*

*Continued*

| Reagent type (species) or resource | Designation | Source or reference | Identifiers | Additional information |
|---|---|---|---|---|
| Chemical compound, drug | Calcium ionophore A23187 | Sigma-Aldrich | Cat# C7522 | 180 ng/ml |
| Chemical compound, drug | CPT-cAMP | Sigma-Aldrich | Cat# C3912 | 50 µM |
| Chemical compound, drug | Prostaglandin E2 | Sigma-Aldrich | Cat# P5640 | 10 µM |
| Chemical compound, drug | Forskolin | Sigma-Aldrich | Cat# F6886 | 10 µM |
| Chemical compound, drug | Pertussis toxin | Sigma-Aldrich | Cat# P7208 | 100 ng/ml |
| Chemical compound, drug | Phorbol 12-myristate 13-acetate | Sigma-Aldrich | Cat# P8139 | 50 ng/ml |
| Chemical compound, drug | Ionomycin | Sigma-Aldrich | Cat# I3909 | 1 µM |
| Chemical compound, drug | Doxycycline | Sigma-Aldrich | Cat# D9891 | 200 ng/ml |
| Chemical compound, drug | CREB inhibitor (666-15) | Millipore Sigma | Cat# 5383 410001 | 1 µM |
| Chemical compound, drug | Cholera toxin | List biological laboratories | Cat# 101B | 1 µg/ml |
| Chemical compound, drug | Ovalbumin | Worthington Biochemical | Cat# LS003054 | 100 µg/ml |
| Chemical compound, drug | MHC class II OVA peptide | GenScript | Cat# RP10610 | 1 µg/ml |
| Chemical compound, drug | HDM extract | Greer Laboratories | Cat# XPB 82D3A2.5 | |
| Chemical compound, drug | Rp-cAMP | BioLog | Cat# A002S | 50 µM |
| Chemical compound, drug | Rolipram | Tocris Bioscience | Cat# 0905 | 10 µM |
| Chemical compound, drug | CE3F4 | Tocris Bioscience | Cat# 4793 | 10 µM |
| Chemical compound, drug | Curdlan | Wako Chemicals | Cat# 034–09901 | 10 µg/ml |

## Mice and cells

Gnas$^{\Delta CD11c}$ mice were produced in our laboratory by crossing of *Gnas* floxed mice (*Gnas*$^{fl/fl}$) with CD11c-Cre mice as described previously (*Lee et al., 2015*). To generate *Irf4*-deficient CD11c cells, *Irf4*$^{fl/fl}$ mice were crossed to CD11c-Cre mice for at least four generations. Cre-mediated GFP expression was used to trace *Irf4* ablation with other markers (CD11c, CD11b and CD8α) in

splenocytes (*Klein et al., 2006*; *Mildner and Jung, 2014*). More than 96% of isolated cDC2s were GFP positive. The Cre⁻*Irf4*^fl/fl littermates were used as controls. IL17A-eGFP mice were bred to OT-II mice to yield IL17A-eGFP/OT-II mice. All mice were kept in a specific pathogen-free (SPF) facility. IL4-eGFP reporter (4Get) mice were originally made by Dr. R Locksley (University of California San Francisco) (*Mohrs et al., 2001*) and were a gift from Dr. M Kronenberg (LAI, San Diego, CA). IL17A-eGFP mice were bred to OT2 mice to yield 4Get/OT2 and IL17A-eGFP/OT2 mice, respectively. All mice were kept in a specific pathogen-free (SPF) facility. *Irf4*-inducible (*Irf4*⁻/⁻) mice (*Ochiai et al., 2013*) were bred by Dr. R Sciammas (University of California, Davis). *Irf5*⁻/⁻ mice were provided by Dr. I R Rifkin (Boston University) (*Takaoka et al., 2005*; *Watkins et al., 2015*). *Il17a*⁻/⁻mice were obtained from Dr. Y Iwakura (The University of Tokyo, Japan) (*Nakae et al., 2002*).

OT-II T cells were isolated by magnetic beads (EasySep Mouse Naïve CD4⁺ T Cell Isolation Kit, StemCell Technologies) from a single cell suspension of splenocytes. Bone marrow (BM) cells were cultured in the presence of GM-CSF (10 ng/ml) for 7 days. Floating cells from the BM culture were applied to FACS-sorting (BD FACSAria II) and CD11c⁺CD135⁺ BM cells (BM-APCs) were used for co-culture with naïve OT-II T cells and transcription factor (TF) analysis by qPCR. cDC2 cells were isolated from single cell suspension of spleens. CD11c⁺CD11b⁺CD8α⁻ splenocytes (*Mildner and Jung, 2014*; *Worbs et al., 2017*) were isolated by FACS sorting and applied to co-culture and TFs analysis. cDC1 were isolated by FACS sorting of CD11c⁺CD11b⁻CD8α⁺ splenocytes and subjected to the same analysis described for cDC2.

## OVA-specific immune responses in the cDC-OT-II co-culture

OVA-specific CD4⁺ T cell response was performed using the DC-OT-II co-culture system as described (*Lee et al., 2015*). Briefly, CD11c⁺CD11b⁺CD8α⁻ splenocyte (cDC2s), CD11c⁺-CD11b⁻CD8α⁺ splenocyte (cDC1s) or CD11c⁺CD135⁺ BM cells were isolated by FACS sorting as described above. cDC2s and cDC1s were loaded with OVA peptide (1 µg/ml) 2 hr before T cell addition and BM-APCs were cultured in complete RPMI 1640 containing OVA protein (100 µg/ml) for 16 hr. Various cAMP agonists indicated in the figures were added in the culture 16 hr before T cell engagement. Specially, curdlan (10 µg/ml) was treated for 24 hr before co-culture. For the IRF4 restoration in Irf4i cDC2s, 200 ng/ml doxycycline was added for 12 hr before co-culture. OT-II T cells were co-cultured with cDC2 ($3 \times 10^5$ cells) at 1:2 ratio in a round-bottom 96-well plate or with BM-APC ($5 \times 10^5$ cells) at 1:1 ratio in a 24-well plate in the serum-free culture medium supplemented with albumin. After 3 days of co-culture, OT-II T cells were stimulated with plate-bound anti-CD3/28 antibodies for 24 hr and then used by ELISA to measure cytokines levels or stimulated with PMA and ionomycin for 3 hr for the T cell lineage marker (qPCR).

## Switch from memory Th2 to Th17 (Fate mapping)

Memory Th2 cells (TEM, CD44⁺CD62L^low or TCM, CD44⁺CD62L^high) were generated by co-culturing of naïve IL-17GFP CD4⁺ OT-II cells with BM-APCs from Gnas^ΔCD11c mice for 3 days and then T1/ST2⁺ cells were sorted by FACS. T1/ST2⁺ cells were used for 2^nd co-culture with OVA peptide (1 µg/ml) loaded-CPT (50 µM) or Cholera toxin (1 µg/ml) treated WT cDC2s. After 3 days of 2^nd co-culture, IL-17GFP CD4⁺ OT-II T cells were stimulated with plate-bound anti-CD3/28 antibodies for 24 hr and then used for ELISA to measure cytokines levels or stimulated with PMA (50 ng/ml) and ionomycin (1 µM) for 6 hr for the GFP expression measurement by FACS.

## Human DC-like cells differentiation

Human myeloid leukemia cell line MUTZ-3 was acquired from Dr. Martin L. Yarmush (Rutgers University School of Engineering, Piscataway, New Jersey). MUTZ-3 cells were maintained and differentiated into DC-like cells as described previously (*Koria et al., 2012*). Briefly, MUTZ-3 cells were cultured in α-MEM (Invitrogen, Carlsbad, CA, USA) supplemented with 20% Fetal bovine serum, 50 µM β-mercaptoethanol and 10% 5637 cell-conditioned media. For generation of DC-like cells, MUTZ-3 cells ($10^5$ cells/ml, 2 ml medium/well) were cultured in growth media supplemented with GM-CSF (100 ng/ml, BioLegend, San Diego), TGF-β (10 ng/ml, R and D systems) and TNF-α (2.5 ng/ml, R and D systems) for 7 days. The human monocytic leukemia cell line, THP-1 was from ATCC. As described previously (*Berges et al., 2005*), THP-1 cell line was maintained in RPMI 1640 supplemented with 10% FCS at a concentration of $2 \times 10^5$ cells/ml. To induce differentiation, rhIL-4 (100

ng/ml) and rhGM-CSF (100 ng/ml) were added. Cells were cultured for 5 days to acquire the DC-like cell phenotype. Medium was exchanged every 2 days with fresh cytokine-supplemented medium. The human leukemia cell line, HL-60 was from ATCC. HL-60 cells were cultured in RPMI 1640 medium containing 10% FCS. For DC-like differentiation, HL-60 cells were incubated in the culture media together with calcium ionophore A23187 (180 ng/ml) and rhGM-CSF (100 ng/ml) for 24 hr, as described (*Yang et al., 2007*). All cell line was regularly tested for mycoplasma contamination by PCR and negative for mycoplasma.

## ELISA measurement of cytokines

Cytokine levels in the supernatant were determined using ELISA kits for IL-4, IL-5, IL-10, IFN-γ, IL-1β, IL-6 and IL-17A (eBioscience, La Jolla, CA) following the manufacturer's instructions.

## Flow cytometry and intracellular staining

The data were acquired by a C6 Accuri flow cytometer (BD Biosciences) and analyzed by FlowJo Software. For the staining of surface molecule, cells were washed with FACS buffer (2% FCS containing PBS) and incubated with the indicated antibodies on ice for 30 min. After two times of wash with FACS buffer, cells were used for analysis. For IRF4, IRF5 and IRF8 intracellular staining, surface marker stained cells were fixed and permeabilized using Cytofix/Cytoperm (BD Biosciences) and stained with Ab for 30 min. After 3 times of wash with permeabilization buffer, the intracellular levels of IRFs were analyzed in the different DC subsets. For the detection of GFP$^+$ cells, harvested CD4$^+$ T cells were stimulated with PMA (50 ng/ml) and ionomycin (1 μM) in the presence of GolgiStop (BD Pharmingen) for 6 hr before FACS analysis.

## Adoptive transfer of cDC2s

Adoptive transfer model was initially described by *Lambrecht et al. (2000)*. Briefly, splenic cDC2s from WT donors were pulsed with HDM (50 μg/ml) or OVA (100 μg/ml), with and without CPT or Curdlan. After 24 hr incubation, the cDC2s were washed with PBS three times, re-suspended in PBS, and transferred into anesthetized WT or IL-17eGFP recipient mice i.n (5 × 10$^5$ cells in 20 μl PBS) on day 0 and 14 (*Lee et al., 2015*; *Machida et al., 2004*). Mice were challenged by 12.5 μg HDM, or 25 μg OVA i.n. on day 19, 20 and 21. One day after the last challenge, mice were analyzed for airway hyper-responsiveness (AHR) to methacholine (MCh). Bronchoalveolar lavage (BAL) fluid was collected for cellular composition (light microscopy) and cytokine analysis. For cytokine analysis from lung tissue, lung single cell suspension was prepared from three lobes from each mouse as described (*Doherty et al., 2011*; *Lee et al., 2015*) and stimulated with HDM (50 μg/ml) or OVA (200 μg/ml) for 3 days. Supernatant from the culture was applied to cytokine analysis by ELISA.

## Quantitative PCR analysis

Isolation of RNA was carried out using an RNA purification Kit (Thermo Fisher Scientific) according to the manufacturer's instructions. The cDNA was synthesized using Superscript III First-Strand system (Invitrogen). Quantitative PCR analysis was performed as described previously (*Lee et al., 2015*). SYBR Green PCR Master Mix was used for real-time PCR (Thermo Fisher Scientific). Samples were run in triplicate and normalized by *Gapdh*. PCR for *Irf5* was performed using Taqman primers according to the manufacturer's instructions. Primer sequences are listed in previous study (*Lee et al., 2015*) and additional sequences are:

*Irf4*: F-AGATTCCAGGTGACTCTGTG, R-CTGCCCTGTCAGAGTATTTC,
*Irf8*: F-CGCTGTAGGAAAAGCAGACC, R-CCTCCAACAACACAGGGAGT,
*Klf4*: F-CTGAACAGCAGGGACTGTCA, R-GTGTGGGTGGCTGTTCTTTT,
*Crem*: F-GCTGAGGCTGATGAAAAACA, R-GCCACACGATTTTCAAGACA,
*Il23a*: F-TCCGTTCCAAGATCCTTCG, R-GAACCTGGGCATCCTTAAGC.

## Expression profiling and genomic analysis

Total RNA was isolated from splenic cDC2 cells using RNeasy columns (Qiagen, Germantown, MD). Non-stranded sequencing libraries were prepared using Illumina TruSeq RNA library kits and sequenced on an Illumina Hi Seq2500 using a SE70 protocol (Illumina, San Diego, CA). ATACSeq was performed on splenic cDC2 cells by the UCSD Center for Epigenomics using a PE75 protocol.

Raw sequenced reads were trimmed for adapter and bar-codes sequences and quality assessed using FastQC (http://www.bioinformatics.babraham.ac.uk/projects/fastqc/). Reads were aligned to the mm10 genome using STAR (*Dobin et al., 2013*) with mgcGene annotations (https://genome.ucsc.edu/cgi-bin/hgTables). Differential expression was determined using DESeq2 (*Anders and Huber, 2010*) in SeqMonk (http://www.bioinformatics.babraham.ac.uk/projects/seqmonk/). ChIPseq reads were identified using the Cistrome Data Browser (http://cistrome.org/db/#/) then raw reads were downloaded from the Sequence Read Archive (https://www.ncbi.nlm.nih.gov/sra) using sra-tools (https://ncbi.github.io/sra-tools/). The reads were aligned with STAR (*Dobin et al., 2013*) then binding peaks identified using the HOMER suite (http://homer.ucsd.edu/). Differential ChIPseq and ATACseq peaks were analyzed using the getDifferentialPeaksReplicates.pl, annotatePeaks.pl and makeMetaGeneProfile.pl scripts, and motif analysis performed using the findMotifsGenome.pl script. The binding-expression predictions were run with BETA (*Wang et al., 2013*). Histone modification co-localization, the transcription factor binding metagene analysis and the de novo motif identification were performed using HOMER. RNAseq, ChIPseq, ATACseq and histone modifications were visualized in IGV (*Robinson et al., 2011*). Heatmaps were generated using heatmap.2 in R. Transcriptional network and pathway enrichment analysis were performed in MetaCore (Genego, Clarivate Analytics, Philadelphia, PA).

## Statistical analysis

Student's t-tests were used to analyze data sets with two groups (GraphPad Prism software). One- or two-way ANOVA were used for multiple groups. All data are represented as mean ± s.e.m unless indicated otherwise. p-values<0.05 were considered significant.

## Study approval

All the experimental procedures were approved by the UCSD-IACUC (#S02240).

## Acknowledgements

We would like to acknowledge assistance from the Genomic and the Histology Shared Resources that are supported by the Moores Cancer Center CCSG Grant (NIH CA023100), UCSD/UCLA Diabetes Research Center Grant (NIH DK063491), Dr. Nissi Varki for her invaluable help in the assessment of mouse pathology and James Lee for assistance with cell preparation. This paper is dedicated to the late Dr. Tim Bigby, a pulmonologist who provided invaluable insights and advice.

## Additional information

### Funding

| Funder | Grant reference number | Author |
| --- | --- | --- |
| Crohn's and Colitis Foundation of America | CCFA327206 | Jihyung Lee |
| National Natural Science Foundation of China | NSFC81500017 | Junyan Zhang |
| National Institute of Allergy and Infectious Diseases | R01AI113145 | Roger Sciammas |
| Eunice Kennedy Shriver National Institute of Child Health and Human Development | HD012303 | Nicholas JG Webster |
| National Cancer Institute | CA196853 | Nicholas JG Webster |
| National Institute of Diabetes and Digestive and Kidney Diseases | DK063491 | Nicholas JG Webster |
| National Heart, Lung, and Blood Institute | HL141999a | Nicholas JG Webster |
| U.S. Department of Veterans Affairs | I01BX000130 | Nicholas JG Webster |

| National Natural Science Foundation of China | NSFC81373128 | Ailin Tao |
|---|---|---|
| Deutsche Forschungsgemeinschaft | NU 53/9-2 | Bernd Nürnberg |
| National Institute of Allergy and Infectious Diseases | AI 070535 | David H Broide |
| National Institute of Allergy and Infectious Diseases | AI113145 | Roger Sciammas |
| Crohn's and Colitis Foundation of America | CCFA330251 | Eyal Raz |
| National Institute of Allergy and Infectious Diseases | AI125860 | Eyal Raz |
| National Heart, Lung, and Blood Institute | HL141999 | Eyal Raz |
| Deutsche Forschungsgemeinschaft | NU 53/13-1 | Bernd Nürnberg |
| National Cancer Institute | CA023100 | Nicholas JG Webster |
| National Institute of Allergy and Infectious Diseases | AI107779 | David H Broide |
| National Institute of Allergy and Infectious Diseases | AI124236 | David H Broide |

The funders had no role in study design, data collection and interpretation, or the decision to submit the work for publication.

### Author contributions

Jihyung Lee, Conceptualization, Data curation, Formal analysis, Funding acquisition, Visualization, Methodology; Junyan Zhang, Young-Jun Chung, Conceptualization, Data curation, Formal analysis; Jun Hwan Kim, Chae Min Kook, Formal analysis; José M González-Navajas, Data curation, Formal analysis; David S Herdman, Formal analysis, Project administration; Bernd Nürnberg, Resources; Paul A Insel, Maripat Corr, Conceptualization, Resources; Ji-Hun Mo, Supervision; Ailin Tao, Data curation; Kei Yasuda, Ian R Rifkin, Resources, Data curation, Methodology; David H Broide, Resources, Data curation, Supervision, Methodology; Roger Sciammas, Conceptualization, Resources, Data curation, Methodology; Nicholas JG Webster, Conceptualization, Resources, Data curation, Formal analysis, Supervision, Funding acquisition, Methodology; Eyal Raz, Conceptualization, Resources, Data curation, Supervision, Funding acquisition, Validation

### Author ORCIDs

Bernd Nürnberg (ID) http://orcid.org/0000-0002-5995-6555
Eyal Raz (ID) https://orcid.org/0000-0001-9320-0260

### Ethics

Animal experimentation: All the experimental procedures were approved by the UCSD-IACUC.

### Decision letter and Author response

Decision letter https://doi.org/10.7554/eLife.49416.sa1
Author response https://doi.org/10.7554/eLife.49416.sa2

## Additional files

### Supplementary files

• Supplementary file 1. Tables of transcriptional profiling (RNAseq). Supplementary Table 1: Genes altered by CPT treatment of cDC2 Table shows genes altered in CPT-treated splenic cDC2 cells. RNAseq data was analyzed by DESeq2 using a FDR < 0.05 multiple testing correction.

Columns indicate gene symbol; chromosome; start and end positions of the gene; chromosome strand; stable Ensembl gene ID; description of gene; mean read counts for CPT-treated WT (CPT), untreated WT (UN), doxycycline-treated *Irf4*^-/- (DOX), *Irf4*^-/- (IRF4KO), and WT littermate (WT) cDC2 cells; fold change for CPT-treated versus untreated (FC); the log2-transformed fold change (log2FC); and the corrected p-value (FDR). Supplementary Table 2: Genes altered in *Irf4*^-/- cDC2 Table shows genes altered in splenic cDC2 cells from *Irf4*^-/- mice. RNAseq data was analyzed by DESeq2 using a FDR < 0.05 multiple testing correction. Columns indicate gene symbol; chromosome; start and end positions of the gene; chromosome strand; stable Ensembl gene ID; description of gene; mean read counts for CPT-treated WT (CPT), untreated WT (UN), doxycycline-treated *Irf4*^-/- (DOX), *Irf4*^-/- (IRF4KO), and WT littermate (WT) cDC2 cells; fold change for CPT-treated versus untreated (FC); the log2-transformed fold change (log2FC); and the corrected p-value (FDR). Supplementary Table 3: Genes altered in IRF4 overexpressing *Irf4*^-/- cDC2 Table shows genes altered in splenic cDC2 cells from *Irf4*^-/- mice that had been treated with doxycycline to over-express IRF4. RNAseq data was analyzed by DESeq2 using a FDR < 0.05 multiple testing correction. Columns indicate gene symbol; chromosome; start and end positions of the gene; chromosome strand; stable Ensembl gene ID; description of gene; mean read counts for CPT-treated WT (CPT), untreated WT (UN), doxycycline-treated *Irf4*^-/- (DOX), *Irf4*^-/- (IRF4KO), and WT littermate (WT) cDC2 cells; fold change for CPT-treated versus untreated (FC); the log2-transformed fold change (log2FC); and the corrected p-value (FDR). Supplementary Table 4: Transcription factor networks derived from CPT-regulated genes. Table shows transcription factor networks generated using genes differentially expressed in CPT-treated cDC2 cells. Networks were generated using GeneGo's MetaCore software. Columns contain network number; transcription factor driving network (Network); gene ontology (GO) processes that are enriched for the network; total number of genes (nodes) in network; number of input differentially-expressed genes (seed nodes) in network; number of canonical pathways in the network; the p-value for the network (p-Value), the z-score (zScore) indicating the number of SDs from the mean for the network, and the z-score corrected for the interactions of non-seed nodes (gScore) for the network. Supplementary Table 5: Transcription factor networks derived from genes differentially expressed in *Irf4*^-/- cDC2. Table shows transcription factor networks generated using genes differentially expressed in *Irf4*^-/- cDC2 cells. Networks were generated using GeneGo's MetaCore software. Columns contain network number; transcription factor driving network (Network); gene ontology (GO) processes that are enriched for the network; total number of genes (nodes) in network; number of input differentially-expressed genes (seed nodes) in network; number of canonical pathways in the network; the p-value for the network (p-Value), the z-score (zScore) indicating the number of SDs from the mean for the network, and the z-score corrected for the interactions of non-seed nodes (gScore) for the network. Supplementary Table 6: Transcription factor networks derived from genes differentially expressed by over-expression of IRF4. Table shows transcription factor networks generated using genes differentially expressed in doxycycline-treated *Irf4*^-/- cDC2 cells. Networks were generated using GeneGo's MetaCore software. Columns contain network number; transcription factor driving network (Network); gene ontology (GO) processes that are enriched for the network; total number of genes (nodes) in network; number of input differentially-expressed genes (seed nodes) in network; number of canonical pathways in the network; the p-value for the network (p-Value), the z-score (zScore) indicating the number of SDs from the mean for the network, and the z-score corrected for the interactions of non-seed nodes (gScore) for the network. Supplementary Table 7: Genes altered in both CPT-treated and *Irf4*^-/- cDC2 Table shows genes differentially expressed in both CPT-treated and from *Irf4*^-/- splenic cDC2 cells. RNAseq data was analyzed by DESeq2 using a FDR < 0.05 multiple testing correction. Columns indicate gene symbol; chromosome; start and end positions of the gene; chromosome strand; stable Ensembl gene ID; mean read counts for CPT-treated WT (CPT), untreated WT (Untreated), doxycycline-treated *Irf4*^-/- (DOX), *Irf4*^-/- (IRF4-KO), and WT littermate (WT) cDC2 cells; the log2-transformed fold change for CPT-treated cDC2 (log2FC CPT/UN); the log2-transformed fold change for doxycycline-treated *Irf4*^-/- cDC2 (log2FC DOX/KO); the log2-transformed fold change for *Irf4*^-/- cDC2 (log2FC KO/WT); and the description of the gene. Supplementary Table 8: Genes altered in both *Irf4*^-/- cDC2 and in the IRF4 over-expressing cDC2 Table shows genes differentially expressed in both *Irf4*^-/- splenic cDC2 cells and in doxycycline-treated *Irf4*^-/- cDC2. RNAseq data was analyzed by DESeq2 using a FDR < 0.05 multiple testing correction. Columns indicate gene symbol; chromosome; start and end positions of the gene; chromosome strand; stable Ensembl gene ID; mean read counts for CPT-treated WT

(CPT), untreated WT (Untreated), doxycycline-treated *Irf4*[-/-] (DOX), *Irf4*[-/-] (IRF4-KO), and WT littermate (WT) cDC2 cells; the log2-transformed fold change for CPT-treated cDC2 (log2FC CPT/UN); the log2-transformed fold change for doxycycline-treated *Irf4*[-/-] cDC2 (log2FC DOX/KO); the log2-transformed fold change for *Irf4*[-/-] cDC2 (log2FC KO/WT); and the description of the gene. Supplementary Table 9: Genes altered in CPT-treated, *Irf4*[-/-] cDC2, and IRF4 over-expressing splenic cDC2 Table shows genes differentially expressed in CPT-treated cDC2, *Irf4*[-/-] cDC2 cells, and *Irf4*[-/-] cDC2 treated with doxycycline to over-express IRF4. RNAseq data was analyzed by DESeq2 using a FDR < 0.05 multiple testing correction. Columns indicate gene symbol; chromosome; start and end positions of the gene; chromosome strand; stable Ensembl gene ID; mean read counts for CPT-treated WT (CPT), untreated WT (Untreated), doxycycline-treated *Irf4*[-/-] (DOX), *Irf4*[-/-] (IRF4-KO), and WT littermate (WT) cDC2 cells; the log2-transformed fold change for CPT-treated cDC2 (log2FC CPT/UN); the log2-transformed fold change for doxycycline-treated *Irf4*[-/-] cDC2 (log2FC DOX/KO); the log2-transformed fold change for *Irf4*[-/-] cDC2 (log2FC KO/WT); and the description of the gene. Supplementary Table 10: Transcription factor networks derived from genes differentially expressed in CPT-treated cDC2, *Irf4*[-/-] cDC2, and in *Irf4*[-/-] cDC2 over-expressing IRF4. Table shows transcription factor networks generated using genes differentially expressed in CPT-treated cDC2, *Irf4*[-/-] cDC2 cells, and *Irf4*[-/-] cDC2 treated with doxycycline to over-express IRF4. Networks were generated using GeneGo's MetaCore software. Columns contain network number; transcription factor driving network (Network); gene ontology (GO) processes that are enriched for the network; total number of genes (nodes) in network; number of input differentially-expressed genes (seed nodes) in network; number of canonical pathways in the network; the p-value for the network (p-Value), the z-score (zScore) indicating the number of SDs from the mean for the network, and the z-score corrected for the interactions of non-seed nodes (gScore) for the network. Supplementary Table 11: Enrichment analysis using the 239 genes differentially expressed in CPT-treated cDC2, *Irf4*[-/-] cDC2, and in *Irf4*[-/-] cDC2 over-expressing IRF4. Table shows the top ten enriched pathways, process networks, and GO processes transcription factor networks generated using genes differentially expressed in CPT-treated cDC2, *Irf4*[-/-] cDC2 cells, and *Irf4*[-/-] cDC2 treated with doxycycline to over-express IRF4. Enrichment was performed using GeneGo's MetaCore software. Columns contain the pathway map, network, or process name; the total number of genes involved; the p-value (p-Value); the FDR corrected p-value (FDR); the number of differentially expressed genes in the pathway, network or process; and the gene symbols for these genes in the pathway, network, or process. Supplementary Table 12: Common genes encoding secreted proteins. Table shows the 239 differentially expressed genes common to CPT-treatment *Irf4*[-/-] or IRF4 over-expression that are annotated to encode secreted proteins. Columns indicate gene symbol; chromosome; start and end positions of the gene; chromosome strand; stable Ensembl gene ID; description of the gene; mean read counts for CPT-treated WT (CPT), untreated WT (Untreated), doxycycline-treated *Irf4*[-/-] (DOX), *Irf4*[-/-] (IRF4-KO), and WT littermate (WT) cDC2 cells; the log2-transformed fold change for CPT-treated cDC2 (log2FC CPT/UN); the log2-transformed fold change for doxycycline-treated *Irf4*[-/-] cDC2 (log2FC DOX/KO); the log2-transformed fold change for *Irf4*[-/-] cDC2 (log2FC KO/WT); the FDR-corrected p-values (FDR); the Uniprot accession number; gene synonyms; the UniProt info; and the UniProt protein name. Table is separated into two panels for genes showing concordant versus discordant regulation by CPT and IRF4. Supplementary Table 13: Common genes encoding plasma membrane proteins. Table shows the 239 differentially expressed genes common to CPT-treatment *Irf4*[-/-] or IRF4 over-expression that are annotated to encode plasma-membrane proteins. Columns indicate gene symbol; chromosome; start and end positions of the gene; chromosome strand; stable Ensembl gene ID; description of the gene; mean read counts for CPT-treated WT (CPT), untreated WT (Untreated), doxycycline-treated *Irf4*[-/-] (DOX), *Irf4*[-/-] (IRF4-KO), and WT littermate (WT) cDC2 cells; the log2-transformed fold change for CPT-treated cDC2 (log2FC CPT/UN); the log2-transformed fold change for doxycycline-treated *Irf4*[-/-] cDC2 (log2FC DOX/KO); the log2-transformed fold change for *Irf4*[-/-] cDC2 (log2FC KO/WT); the FDR-corrected p-values (FDR); the Uniprot accession number; gene synonyms; the UniProt info; and the UniProt protein name. Table is separated into two panels for genes showing concordant versus discordant regulation by CPT and IRF4. Supplementary Table 14: Chromatin accessibility by ATACseq. Table shows the genome-wide ATACseq peak enrichment for CPT-treated cDC2, *Irf4*[-/-] cDC2, or IRF4 over-expressing *Irf4*[-/-] cDC2. Sequence reads were analyzed using HOMER. Columns indicate peak-ID; chromosome; start and end positions of the peak; chromosome strand; peak score; the focus ratio for the region size; the location of the peak; the distance to

nearest transcription start site (TSS); the nearest promoter ID; the Entrez ID; the nearest Unigene ID; the nearest RefSeq gene; the nearest Ensembl gene ID; the gene symbol; gene aliases; the description of the gene; the type of gene; the sequence tag counts; and the peak p-value and adjusted p-value. Table is split into three panels for CPT-treated, $Irf4^{-/-}$ enriched, or IRF4 over-expression enriched peaks. Supplementary Table 15: ChIPseq and ATACseq datasets used in this paper. Table shows ChIP target, SRA file numbers, cells used to generate data, and publication reference. Supplementary Table 16: IRF4-super enhancers in BMDC. Table shows the IRF4 super enhancers derived from IRF4 ChIPseq data. Sequence reads were analyzed using HOMER. Columns indicate peak-ID; chromosome; start and end positions of the peak; chromosome strand; peak score; the focus ratio for the region size; the location of the peak; the distance to nearest transcription start site (TSS); the nearest promoter ID; the Entrez ID; the nearest Unigene ID; the nearest RefSeq gene; the nearest Ensembl gene ID; the gene symbol; gene aliases; the type of gene; and the description of the gene.

- Transparent reporting form

## Data availability

All data supporting the findings of this study are available within the paper or in the supplementary materials.

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
