## [Decision Letter]

**Decision letter after peer review:**

Thank you for submitting your article "Inhibition of IRF4 in dendritic cells by PRR-independent and -dependent signals inhibit Th2 and promote Th17 responses" for consideration by *eLife*. Your article has been reviewed by Tadatsugu Taniguchi as the Senior Editor, a Reviewing Editor, and two reviewers. The reviewers have opted to remain anonymous.

The reviewers have discussed the reviews with one another and the Reviewing Editor has drafted this decision to help you prepare a revised submission.

Reviewer #1:

This manuscript reveals the interaction of cAMP signaling and IRF4 in DC function, and the heterogeneity of cDC2 cells. The results are of interest, but there are issues for the authors to address to improve the overall impact.

1) One of the most critical issues is the causative relationship between cAMP and IRF4. The current data suggest that the effects of cAMP activation and IRF4 deletion are highly similar and phenocopy each other, but such data are correlative. The authors need to establish whether IRF4 truly mediates cAMP signaling, e.g. by expressing IRF4 in cAMP-treated DCs.

2) It is somewhat surprising that ATAC-Seq experiments did not reveal major effects on the selective genes. The authors should do a global analysis to test the extent of altered chromatin accessibility, and the enrichments of transcription factors. Also, conclusions derived from RNA-Seq data are inferred, and additional data should be included to validate these results.

3) The authors need to examine IRF4 expression in a more detailed way, e.g. by flow cytometry and western blot analysis.

4) The paper is difficult to read, and the authors should revise it for better clarity. They should include a model to highlight the major findings. Also, the Discussion section is too long and should be streamlined.

The authors should include the number of independent experiments in the figure legends.

As recommended, we included the number of independent experiments in each figure legends.

*Reviewer #2:*

The authors present compelling data examining the molecular mechanisms underlying the Th2 and Th17 inducing properties of cDC2s. This work is of great relevance to the field, as we continue to developmentally and functionally define cDC lineages. The authors show that PPR-independent pathways involving cAMP-PKA-CREB signaling as well as a PRR-dependent pathway repress IRF4 and KLF4 and lead to pro-Th17 priming phenotype of cDC2. The authors present RNA-seq, ChIP-seq and ATAC-Seq data that advances our understanding of the function of IRF4 generally as well as how cAMP ultimately leads to pro-Th17 gene expression changes. Finally, the authors demonstrate functional relevance to their in vitro findings by adoptive transfer of CPT and Curdlan treated cDC2 within an HDM model, which revealed a Th17 bias and neutrophilic infiltration. While the data presented here is quite complete, I do have some suggestions that would strengthen the claims the authors are making.

1) The IRF4 FACS staining is not uniform throughout the paper. In some cases, it appears that the cDC2 population has a bimodal expression pattern of IRF4 with two peaks on the histogram, while others there is a slight shift. The authors should include a cDC1 control in all histogram plots, which should be low but not negative for IRF4 in order to appreciate the true IRF4 positivity of the cDC2 population. In addition, perhaps MFI would be a better way to measure IRF4 shifts in cDC2 since they should be homogenous as a population.

2) In regards to Figure 1I-M, gating for Th2EM cells should be shown and T1/ST2^+^ cells should be described more thoroughly. Have the authors excluded the possibility that these cells are T regulatory cells? FoxP3^+^ cells have also been described to express ST2. What happens to Foxp3 levels in T1/ST2 OT2 cells with co-culture of CPT- or CT-treated cDC2s?

3) In regards to data collected on BM-APCS, it is well established that DC progenitors in the bone marrow have lineage commitment towards pDC, cDC1, and cDC2 identity. If there is a pro-Th17 effect of CPT on CD135^+^ CD11c^+^ BM-DC, which contains some pre-DC population, this may suggest that there would be subset skewing in mice that lack cAMP production within DC. Thus, do the GNASΔCD11c mice have any subset skewing in peripheral splenic DC populations, i.e. what are the percentages of cDC1, cDC2, and pDC in these mice?

4) Do curdlan or cAMP signaling promote Notch2 expression in cDC2? This could be another factor in addition to IRF5 that promotes the pro-Th17 properties of cDC2.

[Editors' note: further revisions were suggested prior to acceptance, as described below.]

Thank you for submitting your article "Inhibition of IRF4 in dendritic cells by PRR-independent and -dependent signals inhibit Th2 and promote Th17 responses" for consideration by *eLife*. Your article has been reviewed by a peer reviewer, and the evaluation has been overseen by a Reviewing Editor and Tadatsugu Taniguchi as the Senior Editor. The reviewers have opted to remain anonymous.

The revisions have addressed most of the reviewer concerns, but there is a remaining point of reviewer 1 that needs to be addressed.

*Reviewer #1:*

The revised manuscript has largely addressed my questions. However, for my original point 4, the presentation of the data in Figure 4—figure supplement 1 is not clear to this reviewer. The authors should double check their data and interpretation, ideally including the control groups (CPT treatment alone, without DOX, appears to be missing), and all relevant statistical comparisons to fully support their conclusions. Otherwise, they should tone down their conclusions.

---

## [Author Response]

Reviewer #1:This manuscript reveals the interaction of cAMP signaling and IRF4 in DC function, and the heterogeneity of cDC2 cells. The results are of interest, but there are issues for the authors to address to improve the overall impact.1) One of the most critical issues is the causative relationship between cAMP and IRF4. The current data suggest that the effects of cAMP activation and IRF4 deletion are highly similar and phenocopy each other, but such data are correlative. The authors need to establish whether IRF4 truly mediates cAMP signaling, e.g. by expressing IRF4 in cAMP-treated DCs.

As the reviewer mentioned, cAMP activation and IRF4 deletion showed a similar phenotype. In the manuscript, we reasoned that increased cAMP inhibits the IRF4 expression, which subsequently induces Th17 differentiation. We introduced several cAMP-specific inhibitors that clarified the relationship between cAMP and IRF4 expression (Figure 1—figure supplement 4).

We further investigated the relationship between IRF4 and cAMP using IRF4 inducible mice on *Irf4*^-/-^ background (*Irf4-*inducible (*Irf4^-/-^*), Irf4i). BM-APC from Irf4i mice were treated with Doxycycline (Dox.) for the reconstitution of IRF4 and then we treated those cells with CPT. IRF4 expression was significantly increased by Dox treatment, but the subsequent CPT treatment did not change the restored IRF4 expression (Author response image 1). We also used these cells for the co-culture with OT-II T cells. IRF4^-/-^ BM-APC induced Th17 differentiation spontaneously and the increased in IL-17 was suppressed by the recovered IRF4 expression by Dox treatment (Author response image 1). In contrast, reconstituted IRF4 expression in BM-APC induced Th2 differentiation (i.e., IL-5, Author response image 1). However, as CPT did not change the restored IRF4 expression, decreased Th17 and increased Th2 differentiation by IRF4 reconstitution were not changed by CPT treatment. These results indicate that sustained expression of IRF4 in DC mitigate the effects of CPT on cell function. We added this data in the Figure 4—figure supplement 1.

**Author response image 1. respfig1:** The impact of sustained IRF4 expression on cAMP induced Th17 bias. (**A**) IRF4 expression in the WT and Irf4i BM-APCs treated with or without doxycycline (Dox, 200 ng/ml) for 16 hour. After the Dox treatment cells were washed and treated with CPT (50 μM) for 48 hour. (**B, C**) IL-17A and IL-5 levels from the re-stimulated OT-II cells co-cultured with BM-APCs from WT and Irf4i mice under the conditions described above. Data are mean ± s.e.m, n=3 in each group; *p<0.05, **#** p<0.05 compared to WT untreated.

2) It is somewhat surprising that ATAC-Seq experiments did not reveal major effects on the selective genes. The authors should do a global analysis to test the extent of altered chromatin accessibility, and the enrichments of transcription factors. Also, conclusions derived from RNA-Seq data are inferred, and additional data should be included to validate these results.

The ATAC-seq analysis of cDC2 that we performed was indeed a global analysis. The complete data is now included in supplementary data and the sequence files uploaded to SRA. The ATACseq data were analyzed using the HOMER suite for differences between CPT treatment of WT cDC2, *Irf4^-/-^* and WT cDC2, and Doxycycline-treated *Irf4*-inducble (*Irf4^-/-^*) cDC2. The CPT-treated WT cDC2 did not show any significant changes in ATACseq peaks; the *Irf4^-/-^* cDC2 gave 1 differential peak; and the IRF4 re-expressed *Irf4*^-/-^ cDC2 gave 9 differential peaks. Hence, we concluded that cAMP signaling and IRF4 do not cause major alterations in chromatin accessibility.

3) The authors need to examine IRF4 expression in a more detailed way, e.g. by flow cytometry and western blot analysis.

In the manuscript we submitted, we already showed the IRF4 expression in cDC2 or BM-APC by using flow cytometry (Figure 1 and Figure 4, Figure 1—figure supplement 3 and Figure 4—figure supplement 1). However, to make the data clear we added MFI value of each group. Furthermore, we check the decreased IRF4 expression by CPT treatment by using western blot (Author response image 2).

**Author response image 2. respfig2:** Decreased IRF4 expression by CPT treatment in WT cDC2. WT cDC2 were treated with CPT (50 μM) for 48 hour and IRF4 were analyzed by western blot.

4) The paper is difficult to read, and the authors should revise it for better clarity. They should include a model to highlight the major findings. Also, the Discussion section is too long and should be streamlined.

As recommended, we generated the graphical summary to highlight the finding (Author response image 3) and added in the Figure 4I. We also rewrote the manuscript to increase clarity and shorten the Discussion section.

**Author response image 3. respfig3:** A schematic diagram of Th2 inhibition and Pro-Th17 induction by PRR-independent and -dependent signals.

The authors should include the number of independent experiments in the figure legends.

As recommended, we included the number of independent experiments in each figure legends.

Reviewer #2:The authors present compelling data examining the molecular mechanisms underlying the Th2 and Th17 inducing properties of cDC2s. This work is of great relevance to the field, as we continue to developmentally and functionally define cDC lineages. The authors show that PPR-independent pathways involving cAMP-PKA-CREB signaling as well as a PRR-dependent pathway repress IRF4 and KLF4 and lead to pro-Th17 priming phenotype of cDC2. The authors present RNA-seq, ChIP-seq and ATAC-Seq data that advances our understanding of the function of IRF4 generally as well as how cAMP ultimately leads to pro-Th17 gene expression changes. Finally, the authors demonstrate functional relevance to their in vitro findings by adoptive transfer of CPT and Curdlan treated cDC2 within an HDM model, which revealed a Th17 bias and neutrophilic infiltration. While the data presented here is quite complete, I do have some suggestions that would strengthen the claims the authors are making.1) The IRF4 FACS staining is not uniform throughout the paper. In some cases, it appears that the cDC2 population has a bimodal expression pattern of IRF4 with two peaks on the histogram, while others there is a slight shift. The authors should include a cDC1 control in all histogram plots, which should be low but not negative for IRF4 in order to appreciate the true IRF4 positivity of the cDC2 population. In addition, perhaps MFI would be a better way to measure IRF4 shifts in cDC2 since they should be homogenous as a population.

Based on the IRF4 FACS staining, BM-APCs showed homogeneous IRF4^+^ population, but there are two populations with different IRF4 expression level in cDC2. It seems like that the reduction of IRF4 by CPT is mainly happening in the IRF4^high^ population (Figure 1H and Figure 1—figure supplement 3). As reviewer recommended, we added cDC1 control in the data in which showed the shift of IRF4 by CPT in cDC2 cells (Author response image 4) and we replaced the previous data in Figure 1H. The reduction of IRF4 by CPT in Gnas^ΔCD11c^ cDC2 was also showed with cDC1 control. Besides, for the clear data presentation, we added MFI value in flow cytometry data.

**Author response image 4. respfig4:** Intracellular staining of IRF4 in WT cDC1 and cDC2s treated with or without CPT for 48 hour.

2) In regards to Figure 1I-M, gating for Th2EM cells should be shown and T1/ST2^+^ cells should be described more thoroughly. Have the authors excluded the possibility that these cells are T regulatory cells? FoxP3^+^ cells have also been described to express ST2. What happens to Foxp3 levels in T1/ST2 OT2 cells with co-culture of CPT- or CT-treated cDC2s?

The T1/ST2 pathway was originally described in Th2 immunity, but recent studies provided evidence for the role of T1/ST2 pathway on the Treg cells (Pastille et al., 2019). We followed the reviewer’s suggestion and confirmed the FOXP3 expression in the T1/ST2^+^ cells. More than 96% of T1/ST2^+^ cells were effector memory T (TEM, CD44^+^ CD62L^low^) and these cells showed less than 1% of FOXP3 expression (Author response image 5). We also analyzed FOXP3 expression in the T cells from the second co-culture with CT- or CPT treated cDC2 and most of T cells were FOXP3 negative (Author response image 5). In our previous study, we also confirmed that co-cultured OT-II T cells with Gnas^ΔCD11c^ BMAPCs did not show any change in the Foxp3 mRNA expression^2^. This data indicates that the cells we differentiated and used for the second co-culture are very unlikely Treg cells. We added this result in the Figure 1—figure supplement 2.

**Author response image 5. respfig5:** FOXP3 expression in T1/ST2^+^ cells. OT-II CD4^+^ T cells were co-cultured with Gnas^ΔCD11c^ BM-APCs to generate memory Th2 cells (first co-culture). From the first co-culture, T1/ST2^+^ cells were FACS sorted and then used for co-culture with WT cDC2s pretreated with or without CPT or Cholera toxin (CT) (second co-culture). (**A**) Memory T cell marker and FOXP3 were analyzed in the first co-cultured T cells. (**B**) FOXP3 were analyzed in the second co-cultured T cells.

3) In regards to data collected on BM-APCS, it is well established that DC progenitors in the bone marrow have lineage commitment towards pDC, cDC1, and cDC2 identity. If there is a pro-Th17 effect of CPT on CD135^+^ CD11c^+^ BM-DC, which contains some pre-DC population, this may suggest that there would be subset skewing in mice that lack cAMP production within DC. Thus, do the GNASΔCD11c mice have any subset skewing in peripheral splenic DC populations, i.e. what are the percentages of cDC1, cDC2, and pDC in these mice?

We followed the reviewer’s suggestion and checked the percentage of cDC1, cDC2, and pDC in GNAS^fl/fl^ and GNAS^ΔCD11c^ mice. The pDC population was similar in both GNAS^fl/fl^ and GNAS^ΔCD11c^ mice, but GNAS^ΔCD11c^ mice showed higher cDC2 and lower cDC1 percentage compared to GNAS^fl/fl^. This data with the higher IRF4 expression in the GNAS^ΔCD11c^ cDC2 (Figure 1—figure supplement 3) supports pro-Th2 phenotype of GNAS^ΔCD11c^ mice which we already reported (Lee et al., 2015).

**Author response image 6. respfig6:** DC composition in the GNAS^fl/fl^ and GNAS^ΔCD11c^ mice. Splenocytes were isolated from GNAS^fl/fl^ and GNAS^ΔCD11c^ mice and used for the analysis for pDCs and cDCs population.

4) Do curdlan or cAMP signaling promote Notch2 expression in cDC2? This could be another factor in addition to IRF5 that promotes the pro-Th17 properties of cDC2.

Recent studies have divided the cDC2 lineage into Th2-inducing (IRF4^+^/KLF4^+^), and Th17-inducing (IRF4^+^/NOTCH2^+^) subpopulations (Tussiwand et al., 2015; Bedoui and Heath, 2015). We followed the reviewer’s suggestion and checked the Notch2 expression on cDC2 after CPT treatment. As a result, the expression of Notch 2 was increased in CPT treated cDC2.

**Author response image 7. respfig7:** Notch 2 expression on cDC2 after CPT treatment. Splenocytes were isolated from WT mice and treated with CPT (50 μM) for 24hour and then Notch 2 expression on cDC2 was analyzed by FACS.

Additional references:

Pastille, E. et al. The IL-33/ST2 pathway shapes the regulatory T cell phenotype to

promote intestinal cancer. Mucosal Immunol 12, 990-1003, doi:10.1038/s41385-019-

0176-y (2019).

[Editors' note: further revisions were suggested prior to acceptance, as described below.]The revisions have addressed most of the reviewer concerns, but there is a remaining point of reviewer 1 that needs to be addressed.Reviewer #1:The revised manuscript has largely addressed my questions. However, for my original point 4, the presentation of the data in Figure 4 —figure supplement 1 is not clear to this reviewer. The authors should double check their data and interpretation, ideally including the control groups (CPT treatment alone, without DOX, appears to be missing), and all relevant statistical comparisons to fully support their conclusions. Otherwise, they should tone down their conclusions.

We agree with reviewer’s comment that our conclusion without appropriate control group is not persuasive. We already had the data which was not included in the previous version. As recommended, we added the result of the control group, CPT without Dox treatment. Unlike in WT BM-ACPs, CPT treatment did not change the expression of IRF4 in IRF4i BM-APCs without Dox (Author response image 8, green-dotted line). We also added IRF4 expression in the WT BM-APCs with Dox and CPT treatment, which is similar with CPT only group (Author response image 8, orange-dotted line). In addition, we added the T cell differentiation results that CPT treatment without Dox induced Th17 and inhibited Th2 in WT but not in IRF4i BM-APCs (Author response image 8). The statistical comparisons were also added to support out conclusion. With the addition of these data, we can suggest that sustained expression of IRF4 in DCs can blunt the effects of cAMP on their function. We updated the Figure 4—figure supplement 1 with new data and did not change the manuscript.

**Author response image 8. respfig8:** Decreased IRF4 expression in BM-APCs promotes pro-Th17 phenotype and sustained IRF4 expression blocks cAMP-induced Th17 bias. (**A**) IRF4 expression in the WT and Irf4i BM-APCs treated with or without doxycycline (Dox, 200 ng/ml) for 16 hour. After the Dox treatment cells were washed and treated with CPT (50 μM) for 48 hour. (**B, C**) IL-17A and IL-5 levels from the re-stimulated OT-II cells co-cultured with BM-APCs from WT and Irf4i mice under the conditions described above. Data are mean ± s.e.m, n=3 in each group; *****p<0.05, **#** p<0.05 compared to WT-untreated.